# Experimental nonclassicality in a causal network without assuming freedom of choice

Emanuele Polino [1], Davide Poderini[1,2], Giovanni Rodari[1], Iris Agresti[1], Alessia Suprano[1], Gonzalo Carvacho[1], Elie Wolfe [3] ✉, Askery Canabarro[2,4], George Moreno [2,5], Giorgio Milani[1], Robert W. Spekkens[3], Rafael Chaves [2,6] ✉ & Fabio Sciarrino [1] ✉

In a Bell experiment, it is natural to seek a causal account of correlations wherein only a common cause acts on the outcomes. For this causal structure, Bell inequality violations can be explained only if causal dependencies are modeled as intrinsically quantum. There also exists a vast landscape of causal structures beyond Bell that can witness nonclassicality, in some cases without even requiring free external inputs. Here, we undertake a photonic experiment realizing one such example: the triangle causal network, consisting of three measurement stations pairwise connected by common causes and no external inputs. To demonstrate the nonclassicality of the data, we adapt and improve three known techniques: (i) a machine-learning-based heuristic test, (ii) a data-seeded inflation technique generating polynomial Bell-type inequalities and (iii) entropic inequalities. The demonstrated experimental and data analysis tools are broadly applicable paving the way for future networks of growing complexity.

Bell's theorem[1], more than any other result, elucidates the manner in which quantum theory necessitates a departure from a classical worldview[2,3]. Recently, it has been realized that it can be understood as a no-go result for providing a satisfactory account of quantum correlations using a classical causal model[4–7]. Under this reframing, violating a Bell inequality can be understood as attesting to the necessity of using an intrinsically quantum notion of a causal model to achieve a causal account of the correlations[5,6,8–13], and thus as witnessing non-classicality. Furthermore, it becomes clear that such an analysis can be generalized to causal structures that are distinct from the Bell scenario[5,14–26].

Such generalizations are highly relevant to the problem of developing quantum technologies. In the context of the Bell scenario

alone, the possibility of witnessing nonclassicality has applications ranging from quantum cryptography[27] to self-testing[28] and communication complexity problems[29], as well as device-independent information processing[30,31], where the processing can be accomplished while relaxing what needs to be known about the inner workings of the devices. Given that tasks such as these are also of interest in arbitrary quantum networks[32–34], which can have complex topologies, it is evident that there is a need for new data-analysis tools appropriate for witnessing nonclassicality in generic causal structures (see review in ref. [25]). Moreover, so far, all the demonstrations of quantum non-locality, in the Bell scenario (Fig. 1(a)) or in complex networks, relied on the use of external inputs, variables whose values can be freely chosen by the experimenter and which serve to switch between different

[1]Dipartimento di Fisica-Sapienza Università di Roma, P.le Aldo Moro 5, I-00185 Roma, Italy. [2]International Institute of Physics, Federal University of Rio Grande do Norte, 59078-970, P. O. Box 1613 Natal, Brazil. [3]Perimeter Institute for Theoretical Physics, 31 Caroline St. N, Waterloo, ON N2L 2Y5, Canada. [4]Grupo de Física da Matéria Condensada, Núcleo de Ciências Exatas-NCEx, Campus Arapiraca, Universidade Federal de ALagoas, 57309-005 Arapiraca, Alagoas, Brazil. [5]Departamento de Computação, Universidade Federal Rural de Pernambuco, 52171-900 Recife, Pernambuco, Brazil. [6]School of Science and Technology, Federal University of Rio Grande do Norte, Natal, Brazil. ✉e-mail: ewolfe@perimeterinstitute.ca; rafael.chaves@ufrn.br; fabio.sciarrino@uniroma1.it

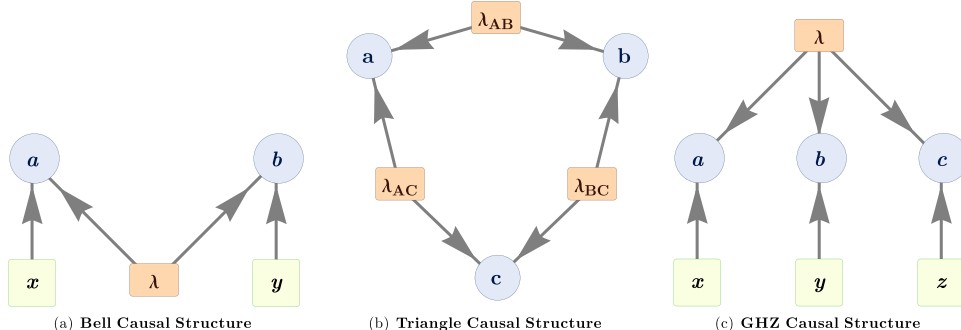

**Fig. 1 | Directed acyclic graph (DAG) representation of different classical causal scenarios. a** The Bell scenario is a causal structure in which a source $\lambda$ correlates the two parties having measurement outcomes $a$ and $b$ and choices $x$ and $y$, respectively. **b** The triangle scenario involves three independent sources $\lambda_{AB}$, $\lambda_{BC}$, and $\lambda_{AC}$, which establish correlations between pairwise stations $A$, $B$, and $C$. Note that measurements in the triangle scenario do not depend on external inputs. **c** The tripartite Bell scenario is also known as the GHZ scenario, after the theorists who identified a nonlocal game for this scenario, which quantum theory predicts can be won with 100% probability.

measurement settings[35–39]. The free choice of measurements lies at the basis of Bell's theorem[40] and in experimental demonstrations, this freedom has to be assumed, or at best made be as plausible as possible[41,42]. By contrast, quantum networks with several independent sources allow the demonstration of nonclassicality without the need for external freely chosen inputs, replacing the freedom of choice assumption with the assumption of independence of the sources[5,19,43–45].

In spite of its significance, this challenge remains largely unexplored, especially from the experimental perspective. This work is a contribution to this effort. We undertake the experimental investigation of a causal structure that has attracted growing attention[5,15,16,18,19,22,23,43,46–55]: the "triangle scenario", depicted in Fig. 1(b). Here, three distant parties each receives a share from two out of three independent sources, and in stark contrast to the Bell scenario, each party implements a single measurement on the systems in its lab, rather than having the freedom to choose among a set of incompatible measurements.

Using a versatile photonic setup with three independent sources (one sharing entanglement and two sharing classical correlations) and the feed-forward of classical information by means of fast optical switches, we provide the first experimental demonstration of classically unrealizable correlations in the triangle structure without the use of external inputs. Importantly, witnessing nonclassicality in this new kind of causal structure goes beyond the standard Bell inequality violation and requires a radically different approach. In the course of doing so, we have enhanced some of the existing tools for testing nonclassicality in generic causal structures both from the experimental and the theoretical perspectives. These enhancements are in the service of making the tools applicable to generic causal structures and arbitrary data, thus paving the way for future experiments in causal networks of growing size and complexity.

## Results

### Beyond Bell's theorem

Leveraging Bell's theorem, Fritz[5] showed the existence of a distribution in the triangle scenario that is realizable quantumly but not classically. Fritz's result is best understood as a quantum no-go theorem akin to Bell's 1964 no-go theorem[1] or the tripartite Greenberger-Horn-Zeilinger (GHZ) argument[56]. As with the distributions described in those works, Fritz's distribution has the feature that certain variables are perfectly correlated, something that is predicted by quantum theory to be possible in principle, but which can never be realized in a real experiment given the unavoidable presence of noise.

It was Clauser, Horne, Shimony, and Holt (CHSH) who first demonstrated how to turn Bell's argument into an experimental test,

by deriving noise-robust inequalities[57]. Similarly, in the tripartite Bell scenario (Fig. 1(c)), the step from the GHZ argument to the possibility of a noise-robust test was achieved by Mermin's inequality[58]. In the case of the triangle scenario, classical causal compatibility inequalities have also been derived[48] but these unfortunately require a degree of sensitivity higher than can reasonably be achieved in current experimental tests. Note that the inequalities derived in ref. [51], by contrast, are not noise-robust because they apply only to distributions exhibiting perfect correlations between certain variables, analogously to Bell's 1964 inequality. New techniques are therefore required to witness nonclassicality in the triangle scenario for the sort of experimental data achievable at present.

Developing new data-analysis techniques is also motivated by considerations of utility. If all one seeks to do is to demonstrate the existence of nonclassicality in a given causal structure, then it is clearly sufficient to implement a dedicated experiment that targets a specific distribution and to test an inequality that is known to be able to witness nonclassicality for the targeted distribution. If, on the other hand, one seeks to use nonclassicality in a given causal structure as a resource for various information-processing tasks, then it is clearly of greater utility to have a test that is able to witness nonclassicality for any distribution that is not classically realizable in the given causal structure.

In some cases, this higher bar can be met by determining all of the classical causal compatibility inequalities associated to a given causal structure and testing for violations of any of these[2]. Unfortunately, however, such a complete characterization soon becomes out of reach, even for seemingly simple scenarios[2,59]. In order to be able to witness nonclassicality on arbitrary data, therefore, it is better to seek a "satisfiability" algorithm, which takes as its input a concrete example of data, and answers the question of classical realizability for that data alone, and in the case of a negative answer, identifies an inequality that is optimized for witnessing its nonclassicality.

We here propose a data-seeded algorithm of this sort that can be used for a generic causal structure. This is achieved by leveraging the fact that the inflation technique for causal inference[18] can reduce the satisfiability problem to a linear program. We also pursue a second route to witnessing nonclassicality on generic data. In this approach, one foregoes deriving inequalities altogether and one simply performs a statistical hypothesis test where the hypothesis is the compatibility of the data with a classical causal model for the given causal structure. Specifically, one implements a variation of the parameters of the model—some of which make explicit reference to the hidden (i.e., unobserved) variables—to try and find the best fit to the data, and one considers the hypothesis falsified at some level of confidence when no good fit can be found. We here show that such hypothesis testing on experimental data can be made feasible for causal networks using the

machine-learning technique developed in ref. [52] where the topology of the causal network is mapped to the topology of a neural network. Finally, suitably mapping the triangle network to a generalization of Bell's scenario that incorporates the possibility of measurement dependence (i.e., that abandons the free choice assumption), we also witness the nonclassicality of the data by using an entropic approach, recently introduced in ref. [44].

Note that for the triangle scenario, our goal is to witness non-classicality of the experimentally realized distribution assuming only that the causal relations among the three measurement nodes and sources are those described by the triangle scenario. If one were to avail oneself of additional assumptions, in particular, assumptions regarding the causal relations among variables within a given laboratory, then one could witness nonclassicality of our experimental data using standard Bell inequalities. Since such additional assumptions do not hold for all setups that can realize a distribution exhibiting a quantum-classical gap, an analysis, which leveraged these additional causal assumptions would not achieve the goal of being applicable to arbitrary data.

### The causal modeling perspective on Bell's theorem

Bell's theorem can be seen as a particular instance of a causal inference problem where for a given hypothesis about the causal structure of the experiment, one inquires whether a classical causal model is able to reproduce the observations[4,6]. In a Bell experiment, a source distributes physical systems between two distant observers—Alice and Bob—they choose the values of their setting variables, denoted by $x$ and $y$, respectively (these determine which of a set of incompatible measurements is implemented at each lab), and then they register the outcomes, denoted by $a$ and $b$, respectively. For simplicity here, we represent the variables and their values with the same letter. The natural causal structure to hypothesize in such an experiment is the one depicted in Fig. 1(a), termed the "Bell scenario".

The assumption of a classical causal model implies that the observed distribution can be decomposed as

$$p(a,b|x,y) = \sum_{\lambda} p(\lambda)p(a|x,\lambda)p(b|y,\lambda). \tag{1}$$

This decomposition is familiar in discussions of Bell's theorem as what follows from assuming a hidden variable model satisfying local causality and certain other conditions[60,61], but it can also be understood as a simple consequence of the causal Markov condition[62] under the assumption that the causal structure is that of the Bell scenario[4,6].

In turn, for a quantum causal model, sources of correlations are not copies of a variable $\lambda$ that is probabilistically distributed but rather pairs of systems that are in a joint quantum state $\rho$ (potentially entangled). Similarly, dependencies among nodes are not represented by conditional probabilities such as $p(a|x,\lambda)$ but by the quantum analogs thereof, completely positive and trace preserving (CPTP) maps, which, in the particular case of a measurement, correspond to a positive operator-valued measure (POVM). Operationally, the quantum description is given by Born's rule, implying that

$$p_Q(a,b|x,y) = \mathrm{Tr}\left[\left(M^A_{a|x} \otimes M^B_{b|y}\right)\rho_{AB}\right], \tag{2}$$

where $\{M^A_{a|x}\}_a$ and $\{M^B_{b|y}\}_b$ are POVMs on $A$ and $B$, respectively.

Bell's theorem[1] asserts that the quantum description can lead to an observable distribution that fails to have a classical explanation in terms of the causal model (1).

### The triangle scenario

Among the simplest quantum networks beyond the paradigmatic Bell causal structure is the triangle scenario of Fig. 1(b). It is distinguished from the tripartite Bell scenario (depicted in Fig. 1(c)) by the fact that

the distant parties are not connected by a 3-way source, but by three 2-way sources.

In the triangle scenario, the correlations that admit a classical realization, i.e., those that are compatible with a classical causal model with the structure of Fig. 1(b), can be written as:

$$p(a,b,c) = \sum_{\lambda_{AB},\lambda_{BC},\lambda_{AC}} p(\lambda_{AB})p(\lambda_{BC})p(\lambda_{AC})$$
$$p(a|\lambda_{AB},\lambda_{AC})p(b|\lambda_{AB},\lambda_{BC})p(c|\lambda_{AC},\lambda_{BC}). \tag{3}$$

By contrast, the correlations which admit of a quantum realization in the triangle network are given by

$$p_Q(a,b,c) = \mathrm{Tr}\left(\rho_{AB} \otimes \rho_{AC} \otimes \rho_{BC} \cdot M^A_a \otimes M^B_b \otimes M^C_c\right), \tag{4}$$

where $\rho_{AB}$ denotes the density operator of the state shared between the nodes $\{A, B\}$ (likewise for $\rho_{AC}$ and $\rho_{BC}$), while $\{M^A_a\}_a$ denotes a POVM on the subsystem in station $A$ (similarly for $\{M^B_b\}_b$ and $\{M^C_c\}_c$).

Recently, it has been theoretically and experimentally demonstrated that a quantum triangle network with a setting variable at each station can give rise to nonclassical correlations[63]. This result, however, employs measurement choices for each of the observers. Here, we go a significant step beyond, showing that nonclassical correlations can emerge even without any freedom of choice.

### The Fritz distribution

In Fritz's example[64] of a distribution $p_Q(a, b, c)$ that is not classically realizable, $a, b$ and $c$ are 4-valued variables, each of which is conceptualized as a pair of binary variables, $a = (a_0, a_1)$, $b = (b_0, b_1)$ and $c = (c_0, c_1)$. Moreover, one can decompose the quantum system $A$ as $A = (A_0, A_1)$, where $A_0$ is the subsystem appearing in $\rho_{AC}$ and $A_1$ is the subsystem appearing in $\rho_{AB}$; analogously for $B = (B_0, B_1)$ and $C = (C_0, C_1)$. The example is realized by taking the three POVMs in Eq. (4) to have the following form:

$$M^{C_0 C_1}_{(c_0,c_1)} = M^{C_0}_{c_0} \otimes M^{C_1}_{c_1},$$
$$M^{A_0 A_1}_{(a_0,a_1)} = M^{A_0}_{a_0} \otimes M^{A_1}_{a_1|a_0}, \tag{5}$$
$$M^{B_0 B_1}_{(b_0,b_1)} = M^{B_0}_{b_0} \otimes M^{B_1}_{b_1|b_0},$$

where $\{M^{C_0}_{c_0}\}_{c_0}, \{M^{C_1}_{c_1}\}_{c_1}, \{M^{A_0}_{a_0}\}_{a_0}, \{M^{B_0}_{b_0}\}_{b_0}$ are all measurements of the $\sigma_z$ Pauli observable, $\{M^{A_1}_{a_1|a_0}\}_{a_1}$ corresponds to one of the two Pauli observables among $\{\sigma_x, \sigma_z\}$ depending on the value of $a_0$, and $\{M^{B_1}_{b_1|b_0}\}_{b_1}$ corresponds to one of the two observables among $\{(\sigma_x + \sigma_z)/\sqrt{2}, (\sigma_x - \sigma_z)/\sqrt{2}\}$ depending on the value of $b_0$. In Fritz's description of a genuinely quantum distribution in the triangle scenario, the state $\rho_{AB}$ is taken to be, for example, a singlet state $|\Psi^-\rangle = (|01\rangle - |10\rangle)/\sqrt{2}$; while $\rho_{AC}$ and $\rho_{BC}$ are maximally entangled states $(|00\rangle + |11\rangle)/\sqrt{2}$. However, since all the measurements on $\rho_{AC}$ and $\rho_{BC}$ are of $\sigma_z$, it is sufficient to take these to be a classically correlated state, namely:

$$\Lambda_{AC} = \Lambda_{BC} = (|00\rangle\langle 00| + |11\rangle\langle 11|)/2. \tag{6}$$

As noted in ref. [5], to see that Fritz's distribution is not classically realizable, it suffices to make a connection to a Bell scenario between Alice and Bob. Note that the variables $a_0$ and $b_0$ determine the measurements that are implemented on $A_1$ and $B_1$. In this respect, they are akin to measurement settings $x$ and $y$ in the usual scenario. However, because $a_0$ and $b_0$ are outputs in the triangle scenario, they could in principle depend on the common source between Alice and Bob. In the usual Bell scenario, of course, if the setting variable $x$ (or $y$) is correlated with $\lambda_{AB}$, one cannot derive the Bell inequalities. The assumption that $x$ and $y$ are not correlated with $\lambda_{AB}$ is termed measurement

independence (or freedom of choice) and is a consequence of the hypothesis that the causal structure for the usual Bell scenario is that of Fig. 1(a).

For the Fritz distribution in the triangle scenario, one can still infer that $a_0$ and $\lambda_{AB}$ are uncorrelated, but now this follows from the fact that $a_0$ is perfectly correlated with the outcome $c_0$, which is causally disconnected from $\lambda_{AB}$. Similarly, the lack of correlation between $b_0$ and $\lambda_{AB}$ is inferred from the perfect correlation between $b_0$ and $c_1$ and the fact that $c_1$ is causally disconnected from $\lambda_{AB}$. If one considers the conditional distribution $p(a_1, b_1|a_0, b_0)$ that is obtained by making the appropriate Bayesian inversion on a distribution $p(a, b, c)$ that is classically realizable in the triangle scenario, then given the independence of $a_0$ (and $b_0$) from $\lambda_{AB}$, this conditional distribution should satisfy the standard Bell inequalities. The fact that the measurements in Fritz's example have been chosen to ensure that the conditional $p_Q(a_1, b_1|a_0, b_0)$ violates a standard Bell inequality implies that the distribution $p_Q(a, b, c)$ is not classically realizable in the triangle scenario.

Any experiment that aims to realize the Fritz distribution in the triangle scenario has the goal of realizing the ideal states and measurements specified above, but due to the inevitability of noise, the states and measurements that are actually implemented are necessarily noisy versions of these. This implies that the correlations between $a_0$ and $c_0$ and between $b_0$ and $c_1$ will not be perfect, which in turn blocks the inference from the classical realizability of $p(a, b, c)$ in the triangle scenario to the classical realizability of $p(a_1, b_1|a_0, b_0)$ in the standard Bell scenario. As such, to witness nonclassicality in such an experiment, one must go beyond the techniques that witness nonclassicality in a standard Bell experiment.

It is worth reiterating here a point made in the beginning of subsection "Beyond Bell's theorem", that our goal is to witness nonclassicality using a data-analysis technique that assumes only the causal structure of the triangle scenario. If we associate a laboratory with each of the nodes in the causal structure, then even though our particular experiment involves specific causal relations between systems within the laboratories, the data analysis cannot make use of this extra structure. In other words, we seek a data-analysis technique that can witness nonclassicality without assuming any such extra structure. This is the sort of assumption that is appropriate for the device-independent paradigm, wherein the experimental devices are presumed to be supplied by an adversary. All that is presumed to be guaranteed is that the causal relations among the laboratories are the ones specified by the triangle scenario. If one could avail oneself of the extra structure that is present in the experiment but not part of the description of the triangle scenario, then standard Bell inequalities would be sufficient to witness nonclassicality. For instance, if one could assume that Alice's output $a_0$ was a faithful copy of the classical randomness she shares with Charlie and that Bob's output $b_0$ was a faithful copy of the classical randomness he shares with Charlie, then one could infer that neither $a_0$ nor $b_0$ could depend on $\lambda_{AB}$ and consequently having $p(a_1, b_1|a_0, b_0)$ violate a Bell inequality would be sufficient to witness nonclassicality. As a second example, if one could assume that the pair of variables $c_0$ and $c_1$ that are outputs of Charlie's laboratory are such that $c_0$ depends only on the source shared with Alice and $c_1$ depends only on the source shared with Bob, then the causal structure being assumed is equivalent to a 4-party line-like structure rather than a triangle scenario. In this case, the full set of Bell inequalities for the conditional distribution $p(a, b|c_0, c_1)$ (where $a = (a_0, a_1)$ and $b = (b_0, b_1)$) are the necessary and sufficient conditions for classicality[65].

In order to be able to witness the nonclassicality of our data assuming only the triangle causal structure, therefore, we cannot rely on standard Bell inequalities. This is why we must have recourse to new data-analysis techniques, such as those presented in subsections "Bounding measurement dependence and violating an entropic

inequality for the triangle network", "Violation of a causal compatibility inequality" and "Bounding measurement dependence and violating an entropic inequality for the triangle network".

## Experimental setup

In our experimental implementation, we used the polarization degrees of freedom of a pair of photons as the two qubits distributed by the source shared between $A$ and $B$, with the $\sigma_z$ eigenstates corresponding to the $\{|H\rangle, |V\rangle\}$ basis of linear polarization. We investigated quantum correlations arising in the triangle network where we aim to have the source between $A$ and $B$ prepare the singlet state. Meanwhile, for the source shared by $A$ and $C$ and the source shared by $B$ and $C$, we aim to have these prepare the classically correlated state of Eq. (6).

Recent years have seen the first experimental implementations of causal structures with a number of independent sources[63,66–68]. In our implementation, the pair of photons associated to the source between $A$ and $B$ are at a wavelength of 810 nm, and are generated through spontaneous parametric down-conversion in a ppKTP nonlinear crystal pumped with a 405nm UV CW-laser, placed inside a Sagnac interferometric geometry[69,70], depicted in the box labeled $\rho_{AB}$ in Fig. 2. To implement the classically correlated sources $\Lambda_{AC}$ and $\Lambda_{BC}$, electrical pulses randomly generated by the shot-noise of distant pairs of single-photon detectors are locally split (boxes labeled $\Lambda_{AC}$ and $\Lambda_{BC}$ in Fig. 2); then they are sent to the stations $A, C$ and $B, C$, respectively, by means of 20m-long electrical cables. Detection of such signals gives values for the bits $a_0, b_0, c_0, c_1$.

Note that this electrical signal sets up classical correlations (i.e., shared randomness) between Charlie and Alice (Bob), and this is a faithful implementation of the state in Eq.(6).

Owing to the probabilistic nature of photon generation and random shot-noise events from detectors, justifying the independence of different sources turns out to be very demanding. This is the reason why the first experimental realization of quantum networks[41,71,72] actually involved a single laser source, thereby requiring a device-dependent justification for the supposed independence of the generated quantum states that relies on the knowledge of the inner process of photon generation. Using spatially separated non-synchronized sources, of different natures, enforces the independence of the sources, also having direct applications in quantum communication protocols. Note, however, that the independence of the sources still remains an assumption, considering that this assumption can always be violated by superdeterministic models[73].

To experimentally achieve the implementation of the separable measurement operators as in Eq. (5), the electrical signals arriving at $A$ and $B$ determine the state of ultra-fast optical switches (Nano Speed Ultra-Fast 1x2 by company Photonwares with a switching time equal to ~8ns) that affect the measurements on the photons coming from $\rho_{AB}$. More specifically, based on which one of the two signals arrives in $A$ ($B$) from $\Lambda_{AC}$ ($\Lambda_{BC}$), the switch will send the photon from $\rho_{AB}$ to two fibers connected to the measurement setups implementing the different polarization measurements. The measurement of the photons is performed by polarization controllers defining the measurement basis followed by in-fiber polarizing beam splitters (PBS) and single-photon detectors. Finally, the four detectors in $A$ ($B$) are electronically connected to a time-to-digital converter, located in the measurement station. The signal from the photon counting, together with the signal from source $\Lambda_{AC}$ ($\Lambda_{BC}$) generate the 4-valued outcome $a$ ($b$). Conversely, in station $C$ the 4-valued outcome $c$ is given by the two classical signals from $\Lambda_{AC}$ and $\Lambda_{BC}$. Note that the electronic signals generated by the detectors are sent to three separated time-to-digital converters, one for each measurement station $A, B, C$, and the recorded events are sent for data processing to a computer located outside the laboratory.

We record experimental events by first choosing a small window $w_1 \sim 4.1$ns, to filter in the signals produced simultaneously from the same source $\Lambda_i$. This allows us to account mostly for 2-fold events,

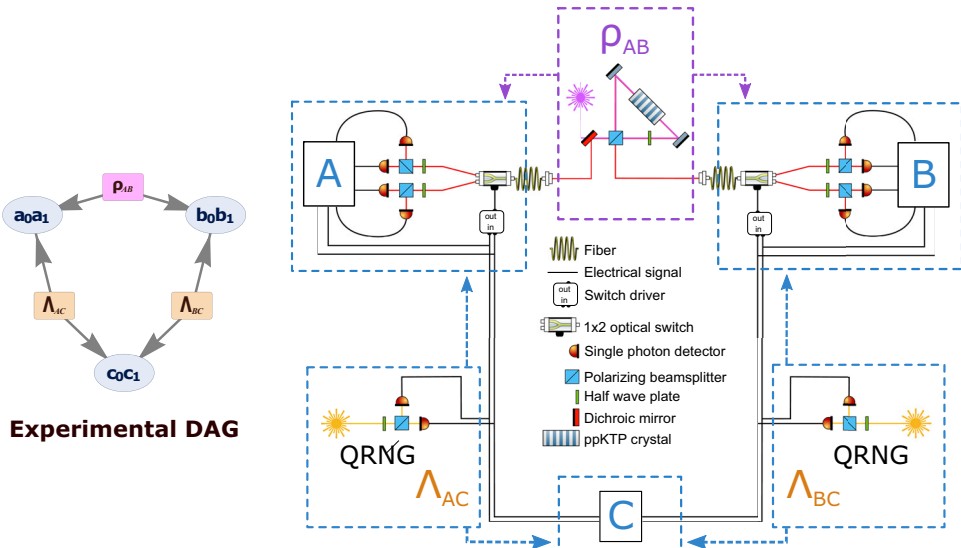

**Fig. 2 | Experimental implementation of the triangle network.** The source $\rho_{AB}$ generates polarization-entangled photon pairs in the singlet state $|\Psi^-\rangle$, by pumping with a continuous wave UV laser a periodically poled potassium titanyl phosphate (ppKTP) crystal. Conversely, $\Lambda_{AC}$ ($\Lambda_{BC}$) produces classically correlated states $(|00\rangle\langle00| + |11\rangle\langle11|)/2$ obtained by splitting the output signal of two single-photon avalanche photodiodes subjected to environmental light noise. In nodes A and B, to implement the measurements needed to reconstruct the probability distribution $p(a, b, c)$, the photons from the source $\rho_{AB}$ are collected by the input single-mode fiber (SMF) of a 5ns rise-time optical switch. In Fritz-like distributions, the measurement result $a_0$ ($b_0$) on part of the source $\Lambda_{AC}$ ($\Lambda_{BC}$) determines the observable to

be measured on the photon coming from $\rho_{AB}$, leading to outcomes $a_1$ ($b_1$). In our implementation, this is achieved by appropriately driving the optical switches through a specially designed electronic driver, which receives signals coming from $\Lambda_{AC}$ ($\Lambda_{BC}$) and drives the output port of the optical switch based on the results $a_0$ ($b_0$). The bit $a_1$ ($b_1$) is obtained by performing a polarization measurement on the photons produced by the ppKTP source through a half-waveplate (HWP) and a polarizing beam splitter (PBS), implemented in fiber. In node C, $c_0$ and $c_1$ are measured independently by directly feeding the electrical signals produced by $\Lambda_{AC}$ and $\Lambda_{BC}$ into a time to digital converter (TDC).

which are due to the same entangled pair, or the same split signal, thus filtering out most of the experimental noise due to the detectors' dark counts and residual environmental light. The 6-fold coincidence events are finally computed by employing a time window equal to $w_2 \sim 20\,\mu s$ inside which an event is defined by the arrival of three two-fold coincidences (see Supplementary Note 1 for more details on data analysis). Such a choice of value for the 6-fold coincidence window represents a compromise between two different requirements. On one side, we want to make such a window as narrow as possible to approximately achieve simultaneity, with respect to both the generation and the measurements, which in principle could lead to an implementation directly addressing the locality loophole. On the other, a broader window is necessary to detect a large enough number of 6-fold coincidences, enhancing the events' rate and thus leading to sufficiently small errors on the measured probabilities in smaller measurement times.

In this demonstration, we do not attempt to achieve space-like separation between the registration of the outcomes $a$, $b$, and $c$. Achieving such a separation would provide the strongest possible justification for the lack of causal influences between the outcomes $a$, $b$ and $c$. It is important to note, however, that it would still not justify the lack of a 3-way common cause.

Furthermore, due to the low efficiencies of the single-photon detectors ($\eta \sim 0.5$) and the fact that the threshold values required for closing the detector loophole in the triangle scenario are not yet known, we rely on the fair-sampling assumption. On this point, we note that even for the much simpler case of the Bell scenario, closing the detector loophole required decades of effort.

**Experimental results**

As stated above, in order to realize the Fritz distribution, it is sufficient to share entanglement only between Alice and Bob's measurement stations, since Alice and Charlie as well as Bob and Charlie can merely share classical correlations. Moreover, using such classical sources (in

our case, a doubled electronic signal) makes it possible to experimentally achieve correlations between Alice and Charlie and between Bob and Charlie that can be almost perfect for the duration of the experiment. Recall that perfect correlation is required for the logic of Fritz's argument to go through, but demonstrating perfect correlations can never be done in an experiment and, importantly, demonstrating nonclassicality in the triangle network in the manner described by Fritz would boil down to violating a standard Bell inequality (sometimes referred to as disguised network nonlocality[74]). So, we did not use this approach here, as it is the goal of our work to introduce and validate data-analysis techniques that would be applicable for any example of a quantum-classical gap in the triangle scenario, including gaps based on distributions that, unlike Fritz's, could be noisy. Figure 3 provides a comparison between the theoretical Fritz distribution reported in panel 3a, obtainable with noiseless states and measurement operators, and the experimentally achieved one reported in panel 3b. The latter one was reconstructed from $\sim 1.4 \cdot 10^6$ events collected in $\sim 10$ h of data taking, achieving a 6-fold coincidence rate of $\sim 38.7$ Hz (see Supplementary Note 2 for the complete distribution).

Even with our approach, employing ultra-fast optical switches and classical correlations shared between $A$ and $C$ and between $B$ and $C$, the measurement outcomes on the state $\Lambda_{AC}$ are not perfectly correlated, nor those on $\Lambda_{BC}$, contrary to the ideal Fritz distribution: specifically, the probability of anti-correlation in each case is found to be $p_{\text{anticorr}} = 3 \cdot 10^{-5}$. As argued, it is the practical impossibility of achieving perfect correlations, which necessitates implementing a hypothesis test for compatibility or a test of causal compatibility inequalities. In what follows, we will focus on three possible avenues: machine-learning techniques[52,75,76], the inflation method[13,18,48,77] and finally, recently derived entropic inequalities[44].

**Excluding the hypothesis of classicality with machine learning**

We follow the approach in[52], the central idea of which is to encode the structure of the causal network under test in the topology of a neural

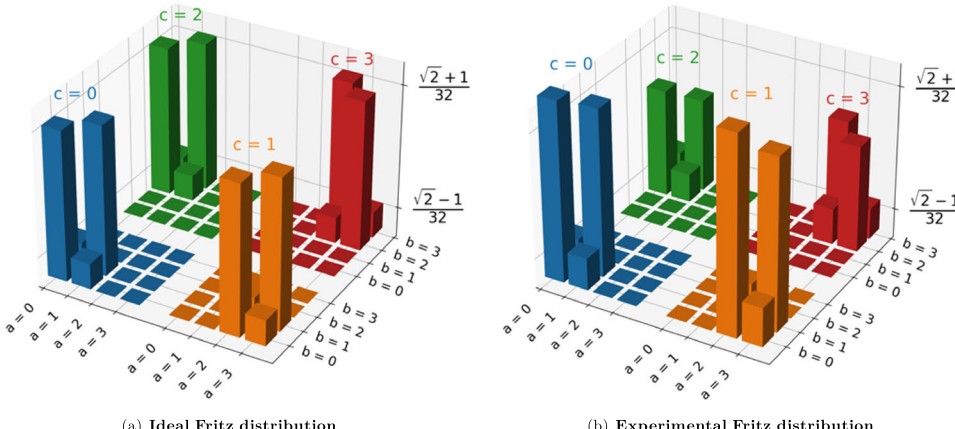

(a) Ideal Fritz distribution

(b) Experimental Fritz distribution

**Fig. 3 | The Fritz distribution, theoretical versus experimental. a** Ideal Fritz distribution computed by choosing $\rho_{AB} = (|HV\rangle - |VH\rangle)/\sqrt{2}$ (a noiseless singlet state); $\Lambda_{AC} = \Lambda_{BC}$ as classically, perfectly correlated mixed states; and the ideal measurement operators described in Eq. (5). **b** Experimental distribution measured in an experimental run. The error bars are calculated using Poissonian statistics and are not visible in the plot. The three indexes $a$, $b$, and $c$ indicate the measurement results, ranging from 0 to 3, corresponding to the three nodes $A$, $B$, and $C$, respectively. The chart bars representing the terms of the probability distribution have different colors based on the value of the outcome $c$.

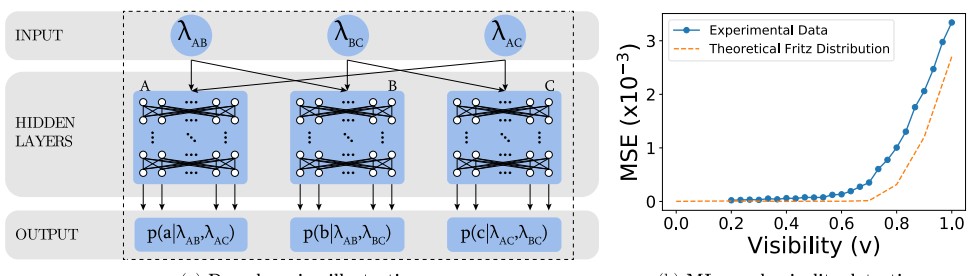

(a) Deep learning illustration.

(b) ML nonclassicality detection.

**Fig. 4 | Neural network for the triangle network. a** Neural network capable of reproducing distributions compatible with the triangle configuration, where number of layers varies from 3 to 6 and number of neurons is either 16 or 32, yielding 8 distinct architectures. The three sources $\lambda_{AB}$, $\lambda_{BC}$, and $\lambda_{AC}$ send information to three parties, Alice, Bob, and Charlie, each receiving, respectively, the pairs $\{\lambda_{AB}, \lambda_{AC}\}$, $\{\lambda_{AB}, \lambda_{BC}\}$, and $\{\lambda_{AC}, \lambda_{BC}\}$. **b** The minimum mean-square error (MSE) distance achieved by the machine as function of the visibility for the experimental data (solid line) and the comparison with the same distance for theoretical Fritz distribution (dashed line). For distinct visibility values, a different ML architecture is the optimum one, strengthening the advantage of using an assembly of oracles. See Methods and Supplementary Note 3 for specific details.

network. Consider the triangle network with quaternary outputs as depicted in Fig. 1(b), where three sources $\lambda_{AB}$, $\lambda_{BC}$, and $\lambda_{AC}$ send information to three parties, Alice, Bob, and Charlie, each receiving, respectively, the pairs $(\lambda_{AB}, \lambda_{AC})$, $(\lambda_{AB}, \lambda_{BC})$, and $(\lambda_{AC}, \lambda_{BC})$, as schematically shown in Fig. 4(a). After locally processing the inputs, they flag a number $a, b, c \in \{0, 1, 2, 3\}$, by sampling the probability distributions $p(a|\lambda_{AB}, \lambda_{AC})$, $p(b|\lambda_{AB}, \lambda_{BC})$ and $p(c|\lambda_{BC}, \lambda_{AC})$, respectively. In the machine-learning algorithm, the input layers to the multilayer perceptrons (MLPs) are composed of the independent uniformly distributed random numbers in the unit interval, i.e., $\lambda_{AB}, \lambda_{BC}, \lambda_{AC} \in [0, 1]$, with the restriction in the flow of information mirroring the causal structure of the triangle network: The $A$-block of the hidden layer receives random numbers $(\lambda_{AB}, \lambda_{AC})$, the $B$-block receives $(\lambda_{AB}, \lambda_{BC})$ and the $C$-block receives $(\lambda_{BC}, \lambda_{AC})$. Therefore, individual inputs belong to $\mathbb{R}^2$ (i.e., they have length 2). For the training, we provide batches of $(N_{batch}, 2)$ dimension for the corresponding MLP for each of the three blocks.

If a certain probability distribution $p(a, b, c)$ is compatible with a classical causal model on the triangle causal structure, then a set of three independent neural networks mimicking the topology of the triangle should be able to reproduce the distribution. By numerically sampling over different values of the random numbers $\lambda_{AB}$, $\lambda_{BC}$ and $\lambda_{AC}$ one can construct the approximation $\tilde{p}(a,b,c)$ by averaging the Cartesian product of the output conditional probabilities corresponding to each party. See Methods for more details.

In turn, if the distribution under test is nonclassical, the neural network will be unable to mimic the distribution perfectly, producing considerable errors. To quantify how much the machine model can approximate the target/experimental distribution, we employ the element-wise mean-square error (MSE), also termed as L2-norm error, between $p(a, b, c)$ and $\tilde{p}(a,b,c)$. This is given by MSE $= \frac{1}{64}|p(a,b,c) - \tilde{p}(a,b,c)|_2$ and can be understood as a measure of nonclassicality[76]. By repeated iterations, the neural network can be optimized in order to minimize this distance, since it should be close to zero if the target distribution has a classical model that the machine manages to approximate. Clearly, however, even if the distribution is compatible with the triangle network, due to numerical precision and the finite size of the neural network, the distance will never be exactly zero. To address this issue, we mix our experimental probability $p(a, b, c)$ with the flat distribution $p_I(a, b, c) = 1/64$, which is compatible with the triangle structure, so that the machine is asked to retrieve the best possible model for the mixed distribution $\tilde{p} = v\,p + (1 - v)p_I$. If $p$ has no classical explanation, then we expect that, as one increases the weight $v$ of $p$ in the mixed distribution $\tilde{p}$, there is a range of values wherein a classical model of $\tilde{p}$ remains possible and MSE is very small, but that there exists a threshold value beyond which MSE begins to increase,

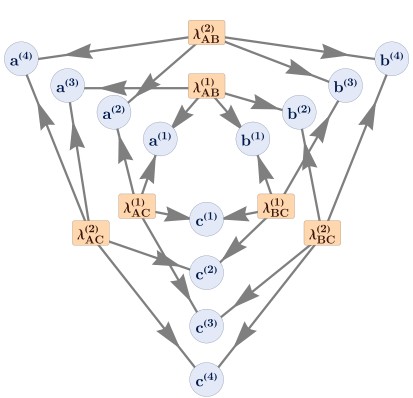

(a) Triangle network second-order inflation

(b) The coefficients of a quadratic inequality

**Fig. 5 | Inflation technique for the triangle network. a** The second order inflation graph of the triangle network. Such an inflation doubles the number of latent variables relative to the triangle scenario, having six latent variables $\{\lambda^{(1)}_{AB}, \lambda^{(2)}_{AB}, \lambda^{(1)}_{BC}, \lambda^{(2)}_{BC}, \lambda^{(1)}_{AC}, \lambda^{(2)}_{AC}\}$. The inflation quadruples the number of observable random variables of the triangle scenario, having twelve observable random variables $\{a^{(1)}, b^{(1)}, c^{(1)}, a^{(2)}, b^{(2)}, c^{(2)}, a^{(3)}, b^{(3)}, c^{(3)}, a^{(4)}, b^{(4)}, c^{(4)}\}$. Distributions compatible with this inflated structure satisfy symmetry properties, and have marginals corresponding to products of triangle-compatible distribution. This can be exploited to derive

suitable causal compatibility inequalities that are violated by the experimental data. **b** This plot depicts the $64 \times 64$ coefficients $y_{a_1 b_1 c_1 a_2 b_2 c_2}$ for a quadratic inequality of the form of Eq. (7) such that the left-hand side is nonnegative on all distributions compatible with the classical triangle scenario, but which evaluates to the negative number $V_{exp} = -0.02436 \pm 0.00016$ on our experimental data. The $x$-axis ranges over the values of $(a_1, b_1, c_1)$ while the $y$-axis ranges over the values of $(a_2, b_2, c_2)$, and the color at a given point denotes the value of $y_{a_1 b_1 c_1 a_2 b_2 c_2}$ according to the mapping set out in the legend.

and the machine cannot make an almost perfect approximation anymore.

As shown in Fig. 4(b), only below a certain threshold value around $v_{crit} = 1/\sqrt{2}$[52], can the machine learn $\tilde{p}$ while it fails to do so for higher values of $v$. This analysis gives a strong indication of the nonclassicality of $p$, but given that there is no guarantee that the machine finds the optimal parameters, it does not guarantee it. To overcome this limitation, in the following we present two alternative techniques.

**Violation of a causal compatibility inequality**

In order to demonstrate the nonclassicality of the experimental data relative to the triangle causal network, we seek to identify some causal inequalities, which must be satisfied by all distributions compatible with the *classical* triangle network but which are violated by our experimental statistics. To this end, we turn to the inflation technique for causal inference introduced in ref. [18].

As detailed in the Methods, the inflation technique relates *compatibility with a given causal structure $\mathcal{G}$* to *feasibility of a linear program (LP)*. If the LP related to an inflation of $\mathcal{G}$ (see Fig. 5(a)) is found to be infeasible, then evidently $p$ is incompatible with $\mathcal{G}$. In our case, $\mathcal{G}$ is taken to be the classical triangle scenario causal structure depicted in Fig. 1(b).

In the case of infeasibility, the algorithm returns an infeasibility witness, in the form of an inequality. In this way, we can find a causal compatibility inequality tailored to the specific experimental data we obtained. Using the second order inflation of the triangle network shown in Fig. 5(a), one can derive causal compatibility inequalities (satisfied by all triangle-compatible $p(a, b, c)$) of the form

$$V \equiv \sum_{\substack{a_1 b_1 c_1 \\ a_2 b_2 c_2}}^{\in \{0,1,2,3\}^{\times 6}} y_{a_1 b_1 c_1 a_2 b_2 c_2} \, p(a_1, b_1, c_1) p(a_2, b_2, c_2) \geq 0, \tag{7}$$

where the $y$ are real coefficients.

As further detailed in the Methods, the LP of the inflation technique may be specially adapted to yield *elegant looking* causal compatibility inequalities; namely, where *sets* of monomials are each associated to a single (i.e., uniform) coefficient. Working with such an *adapted* LP can be orders of magnitude less computationally

demanding as compared to the unadapted LP. However, it may be the case that despite a given distribution leading to infeasibility in the *unadapted* primal LP, there may not exist any inequality *with restricted coefficients* capable of witnessing that fact. As such, one is motivated to carefully select a coefficient restriction, which *matches* the specifically targeted distribution: one should only impose that a pair of monomials should share a uniform coefficient in the inequality if the given distribution would lead to both monomials being evaluated to the same numerical value (within a small tolerance). One cannot impose arbitrary coefficient uniformity restrictions. The Methods contains an explanation for why certain special coefficient restrictions may be justifiable. We employed the *ideal* theoretical Fritz distribution as our guide when selecting our LP adaptation, rendering moot the selection of a numerical tolerance. We stress, however, that a theoretical guide is *not* a prerequisite for optimally adapting the inflation technique LP to witness the nonclassicality of experimental data: it is perfectly possible to isolate the near-symmetries in the experimental data without the educated guess provided by a theoretical model.

The infeasibility witness obtained by the program for our data yields an inequality of the form of Eq. (7), which is violated by the experimental data by several standard deviations: in this way, we unambiguously demonstrate the emergence of nonclassicality in the triangle network, without relying on Bell's theorem. We depict the particular coefficients $y_{a_1 b_1 c_1 a_2 b_2 c_2}$ defining the inequality that we obtained from the adapted LP in Fig. 5(b). Denoting the value that the data gives for the left-hand-side of this inequality by $V_{exp}$, we obtain $V_{exp} = -0.02436 \pm 0.00016$ (using a 6-fold coincidence window $w_2 \sim 20\,\mu s$), corresponding to a violation of the inequality by 152 standard deviations. In Fig. 6, we plot $V_{exp}$ as a function of the choice of the 6-fold coincidence window $w_2$. As expected, by increasing $w_2$, we increase the detection rate of 6-fold events, in turn decreasing the statistical error on the computed value of $V_{exp}$, shown in the figure with the red shadowed area.

**Bounding measurement dependence and violating an entropic inequality for the triangle network**

Another approach that can be used to robustly demonstrate the nonclassicality of the generated data is to map the triangle network into a modification of the Bell scenario, in a similar way to Fritz's

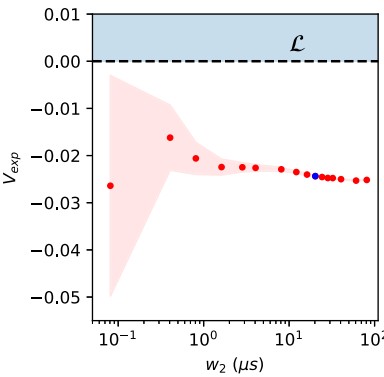

**Fig. 6 | Inflation inequality violation versus 6-fold coincidence window.** Values of violation of the causal compatibility inequality, which has been optimized over the experimental data corresponding to the blue point (a window $w_2 \sim 20\,\mu s$), as a function of the 6-fold coincidence window $w_2$. The red shadowed area represents the statistical error on the computed value of $V_{exp}$, estimated employing Monte Carlo methods. The blue shadowed region $\mathcal{L}$ indicates the values obtainable by a classical causal model.

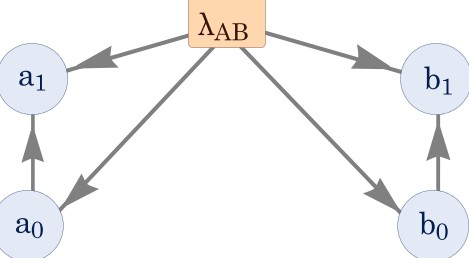

(a) Bell scenario without freedom of choice.

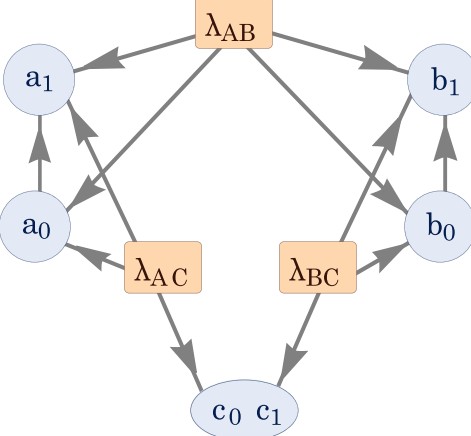

(b) Extended Bell scenario as a triangle network.

**Fig. 7 | Triangle scenario from extended Bell scenario. a** Extended Bell scenario with measurement dependence. Relative to the standard Bell scenario, the source $\lambda_{AB}$ is presumed to influence not only the outcomes $a_1$ and $b_1$, but the setting variables $a_0$ and $b_0$ as well. This allows for measurement dependence and can describe superdeterministic models[73]. **b** Extended Bell causal structure with measurement dependence mapped into the triangle scenario. Relative to the standard Bell scenario, one posits an additional laboratory, associated to Charlie, an additional source $\lambda_{AC}$ between Alice and Charlie and an additional source $\lambda_{BC}$ between Bob and Charlie. Correlations between $a_0$ and $c_0, c_1$ imply an upper bound on the potential dependence of $a_0$ on $\lambda_{AB}$, described by the entropic inequality in Eq. (8). Similarly, correlations between $b_0$ and $c_0, c_1$ imply an upper bound on the potential dependence of $b_0$ on $\lambda_{AB}$.

original proof of nonclassicality in the triangle scenario. In this modification, any amount of measurement dependence is in principle allowed between the hidden variable and the measurement settings. Consequently, even though the scenario is related to Bell's, the nonclassicality exhibited necessarily goes beyond that which one finds in Bell's scenario because in the latter measurement dependence allows for a classical account of any correlations. Indeed, in a Bell scenario where causal influences between the source and the measurement settings are allowed, some amount of measurement independence has to be *assumed* in order to witness nonclassicality from the data[60,78,79], otherwise, any violation of a Bell inequality can be explained by classical local models[80].

In the modified scenario, one can use entropic inequalities to put an upper bound on the amount of measurement dependence, as demonstrated in ref. [44]. The modification can be understood as a two-step departure from Bell's scenario. In the first step, depicted in Fig. 7(a), one allows there to be a common cause not only on the pair of outcome variables, but on all four of the observed variables, meaning that a given outcome variable shares a common cause with the setting variable at the opposite wing; this is a relaxation of the assumption of freedom of choice[41,60,61,81]. In the second step, one introduces additional observed variables $c_0 c_1$ and a variable $\lambda_{AC}$ that is a common cause to Alice's setting and outcome ($a_0, a_1$) and $c_0 c_1$, as well as a variable $\lambda_{BC}$ that is a common cause to Bob's setting and outcome ($b_0, b_1$) and $c_0 c_1$ (see Fig. 7(b)).

Referring to the directed acyclic graph (DAG) of the triangle network shown in Fig. 2, we map the measurement settings of the two stations A and B of the Bell scenario to the variables $a_0$ and $b_0$, and the measurement outcomes are mapped to the variables $a_1$ and $b_1$. It is clear, therefore, that if one lumps $a_0$ and $a_1$ together, and similarly for $b_0$ and $b_1$, the modified Bell scenario can be seen to have the form of the triangle network.

In this modified Bell scenario, shown in Fig. 7(b), one can lower bound the measurement dependence, quantified via the mutual information $I(\lambda_{AB}: a_0, b_0)$ between the source $\lambda_{AB}$ and the measurement settings $a_0$ and $b_0$, relating it with the violation of the CHSH inequality[61,82]. Further, employing the entropic approach[8,46,83,84], this mutual information can also be upper bounded by an entropic function that involves only observable variables and so can be extracted directly from the experimental data. Combining both the upper and lower bounds on $I(\lambda_{AB}: a_0, b_0)$, one arrives at a Bell inequality blending probabilities and entropies, the violation of which witnesses the nonclassicality of the data, irrespectively of any potential measurement

dependence $I(\lambda_{AB}: a_0, b_0)$ present in the experiment. This inequality is given by (see ref. [44] for the further details)

$$\mathcal{E} \equiv 2 - S^{\text{CHSH}} + \sqrt{\frac{16\Theta(a_0, b_0, C)}{\log_2 e}} \geq 0, \tag{8}$$

where $S^{\text{CHSH}}$ is the standard CHSH quantity evaluated on $P(a_1 b_1 | a_0 b_0))$[57], and

$$\Theta(a_0, b_0, C) := \min \begin{cases} H(a_0, b_0 | C), \\ H(a_0, b_0) - I(a_0 : b_0 : C) - I(a_0 : C) - I(b_0 : C), \\ H(a_0, b_0) + H(C) - 2I(a_0 : b_0 : C) - 2I(a_0 : C) - 2I(b_0 : C), \end{cases} \tag{9}$$

with $I(a_0 : b_0 : C) := H(a_0, b_0, C) - H(a_0, b_0) - H(a_0, C) - H(b_0, C) + H(a_0) + H(b_0) + H(C)$ the tripartite mutual information and $H(X) = -\sum_x p(x)\log p(x)$ the Shannon entropy relative to the variable $X$.

Using the experimental data in Fig. 3, we obtain a value $\mathcal{E}_{exp} = -0.340 \pm 0.001$, violating the bound of Eq. (8) by 340 standard deviations and thereby demonstrating nonclassicality.

## Discussion

The triangle scenario has particular novelty as a means of witnessing nonclassicality insofar as there is no known way to obtain classical causal compatibility inequalities for it from standard Bell inequalities. This is in contrast to the two other causal structures distinct from the Bell scenario that have been experimentally investigated previously, namely, the instrumental scenario[24] and the bilocality scenario[66,68,71,72,85,86]. In the case of the instrumental scenario, it suffices to process the Bell inequalities by forcing equality between the value of the setting variable at one wing and the value of the outcome variable at the opposite wing[87]. In the case of the bilocality scenario, by post-selecting on the outcome of the measurement that accesses both sources, the other two measurements can be proven to satisfy the Bell inequalities in a classical causal model (via an analog of entanglement-swapping)[16,17]. Such short-cuts to deriving noise-robust causal compatibility inequalities, however, are not available in the triangle scenario.

Another peculiar aspect of the triangle scenario is the possibility to show new forms of nonclassicality that do not require the use of external inputs freely chosen by the experimenter, but instead rely on the assumption of independence of the sources, as shown by Fritz[5]. In this work, we realized for the first time a triangle network without external inputs, proving the emergence of nonclassicality in this new regime, up to detection and locality loopholes. This has been possible by employing fast feed-forward of measurement in an optical setup comprising an entangled photon source and two sources of classical correlations. In order to demonstrate the nonclassicality of the experimental data, we had to extend pre-existing data-analysis techniques, making them suitable to detect nonclassicality in noisy distributions.

The data-analysis techniques we have presented here are also distinguished insofar as they have the capacity to witness non-classicality for any distribution that might arise in an experiment, whereas previous experiments witnessing nonclassicality in causal structures beyond Bell have used tools that can only witness the nonclassicality of limited classes of target distributions. This approach thus extends data-seeded techniques previously limited to the standard bipartite Bell scenario[2,3,88,89] to the realm of more complex causal networks.

The employed data-analysis techniques and aspects of our photonic setup provide a scalable platform in which nonclassicality can be witnessed in networks of growing size and of arbitrary topology. In particular, the implemented measurements are based on local wirings, i.e., separable measurements with classical feedbacks, making the approach scalable. Furthermore, it is widely speculated that the triangle scenario may admit distributions, which imply a no-go result whose logic is entirely independent of that of Bell's theorem[19,90]. These are likely to require entangled measurements as well as three sources of entanglement, and consequently, integrating such measurements and sources into our set-up may open the way to experimentally targeting distributions, which are thought to exhibit these new types of nonclassicality.

Finally, this work can also pave the way for future applications in quantum communications involving several sources and measurement stations.

## Methods

### Details on the machine-learning implementation

The number of samples we sum over, i.e., the batch size, is $N_{batch} = 10,000$. We decided to vary the architecture of the neural network using different number of layers ($n_{layers} = [3, 4, 5, 6]$) and number of neurons ($n_{neurons} = [16, 32]$), accounting for an assembly of 8 neural networks independently trained, in order to obtain better approximations by taking the minimum or the average of the predictions. The ensemble of networks also reduces the probability of being trapped in optimization local minima and enhances the relative expressive power of the method in comparison to a single architecture; see Fig. 4(a). As pointed out in ref. [52], ideal values for parameters and hyper-parameters vary for distinct triangle scenarios, therefore the strength of the ensemble approach also varies. The reader is referred to the Supplementary Note 3 for more specific details.

### Details on the inflation technique

At its core, the inflation technique at $n^{th}$ order shows that

- IF: A distribution $p$ is compatible with a given classical causal structure $\mathcal{G}$
- THEN: For the $n^{th}$ order inflation graph $\mathcal{G}'$ induced by $\mathcal{G}$ there must exist some larger distribution $p'$ pertaining to the observable nodes in $\mathcal{G}'$ such that

1. $p'$ possesses certain symmetry properties related to automorphisms of $\mathcal{G}'$, and
2. the distribution $p^{\otimes n}$—defined as $n$ identical but independently distributed (I.I.D.) copies of $p$—arises as a marginal distribution of $p'$. These conditions implicitly define a linear program (LP). In the Supplementary Note 4, we elaborate on the required marginal symmetry properties, which must be satisfied by distributions compatible with the second order inflation graph depicted in Fig. 5(a).

Farkas' duality lemma tells us how to extract a *certificate of infeasibility* whenever a LP is infeasible[91]. Note that Farkas' lemma applies to convex optimization in general[92]; linear programming is just a special case. For the primal LP defined by second order inflation, the certificate of infeasibility is a dual vector $\mathbf{y}$ such that $\mathbf{y} \cdot p^{\otimes 2} \geq 0$ holds for all instances of $p^{\otimes 2}$, which make the primal LP feasible. Given such a dual vector $\mathbf{y}$, one certifies the infeasibility of $p^{\otimes 2}_{nonclassical}$—i.e., one certifies the *incompatibility* of $p_{nonclassical}$ with a classical causal model with the structure $\mathcal{G}$—whenever one finds that $\mathbf{y} \cdot p^{\otimes 2}_{nonclassical} < 0$. Hence, the certificate $\mathbf{y}$ yields a quadratic polynomial inequality satisfied by all distributions $p$, which are compatible with $\mathcal{G}$.

We employed the "hierarchy" version of inflation defined in ref. [77] due to its computationally efficient and data-agnostic implementation.

The second order inflation graph of the classical triangle network is depicted in Fig. 5(a), and the $p'$, which is posited to exist would pertain to the twelve observable random variables depicted in Fig. 5(a), namely $\{a^{(1)}, b^{(1)}, c^{(1)}, a^{(2)}, b^{(2)}, c^{(2)}, a^{(3)}, b^{(3)}, c^{(3)}, a^{(4)}, b^{(4)}, c^{(4)}\}$.

The LP implied by inflation is as follows. The condition for the existence of $p'$ can be understood as a collection of very many inequality constraints (every probability, which makes up $p'$ must be nonnegative) along with one equality constraint (the sum of all probabilities comprising $p'$ totals unity). The symmetry requirements of $p'$ can be understood as equality constraints relating the various probabilities comprising $p'$. Finally, the requirement that $p^{\otimes 2}$ is a marginal of $p'$ can be understood as equating $p^{\otimes 2}$ evaluated at a particular set of values for its arguments to a sum over all those probabilities of $p'$, which agree on these values. In other words, if $p$ is compatible with $\mathcal{G}$, then some collection of equality and inequality constraints are simultaneously satisfiable; i.e., some LP should be feasible.

The Farkas infeasibility certificate of the LP defined by inflation constitutes quadratic inequalities, which are satisfied by all triangle-compatible distributions but violated by the nonclassical distribution whose triangle-incompatibility is witnessed by inflation. See Supplementary Note 4 for an explicit walk-through of the inflation technique in full detail.

### Adapting polytope membership LPs to yield symmetric inequalities

It can be insightful to compare the LP defined by inflation to the more familiar LP associated with Bell nonlocality. In Bell nonlocality, a family of conditional probability distributions (a.k.a. a "correlation") is said to

admit a local hidden variable model (LHVM) if and only if corresponding *vector of all conditional probabilities* lies within the local polytope. When a correlation does not admit a LHVM explanation, then we can always find a separating hyperplane (typically a facet of the local polytope) such that the vector of conditional probabilities associated with the given correlation lies strictly to one side of the hyperplane whereas all LHVM-explainable correlations correspond to vectors of conditional probabilities in or on the other side of the hyperplane. Thus, hyperplanes that distinguish all LHVM-explainable vectors from some other are equivalent to Bell inequalities; these hyperplanes, which correspond to facets of the local polytope are equivalent to *facet-defining* Bell inequalities.

The picture is quite similar when thinking about the LP associated with inflation. Instead of vectors of conditional probabilities, however, we are considering vectors whose elements are products of unconditional probabilities, i.e., vectors of probability monomials. The LP of inflation similarly defines a polytope: a vector of monomials is in the polytope iff the primal LP is feasible; the objective of the dual LP is to return a separating hyperplane such that

1. the given vector of monomials is as far from the hyperplane as possible, and
2. such that all vectors, which would make the primal LP feasible lie on or on the other side of the hyperplane.

Without loss of generality, a polytope may be defined in terms of its extremal points. Let $M^{d,n}$ be a $d \times n$ matrix whose $n$ columns correspond to the extremal points of the polytope, each of which is a vector in dimension $d$, and where we have introduced a notation of marking an object's dimension in superscript for pedagogical clarity in what follows. A vector $v^d$ lies withing the polytope (technically, the LP formulations here apply to both bounded polytopes and unbounded polycones) if and only if

$$\begin{aligned} \text{there exists some} \quad & x^n \\ \text{such that} \quad & M^{d,n} \cdot x^n = v^d, \\ \text{where} \quad & x^n \geq \mathbb{0}^n. \end{aligned} \tag{10}$$

We can relax the *satisfiability* LP of Eq. (10) into an *optimization* problem, which measures the *degree of primal infeasibility*. One natural measure of the infeasibility of Eq. (10) is defined by the following optimization problem:

$$\begin{aligned} \max_{x^n, s^n} \quad & -\mathbb{1}^n \cdot s^n \\ \text{such that} \quad & M^{d,n} \cdot (x^n - s^n) = v^d, \\ \text{where} \quad & x^n \geq \mathbb{0}^n \quad \text{and} \quad s^n \geq \mathbb{0}^n. \end{aligned} \tag{11}$$

Note that if the LP of Eq. (10) can be satisfied, then the objective in Eq. (11) can be reach up to 0; conversely, if the objective in Eq. (11) is strictly negative over all variables, which satisfy that LP's conditions, then the LP in Eq. (10) is evidently infeasible. The formal dual to the above LP can then be used to extract optimal separating hyperplanes. The astute reader may notice that even the reformulated LP as given in Eq. (11) may not always be feasible; it can only be satisfied if $v^d$ is wholly in the *linear span* of the columns of $M^{d,n}$. If $v^d$ has some component orthogonal to that linear span, then the primal formulation in Eq. (11) is infeasible and the dual formulation in Eq. (12) is unbounded. See Appendix B of ref. [93] for alternative relaxations of an LP satisfiability problem into an optimization problem, and the connection therein to distance measures such as robustness and nonlocal fraction. Namely,

$$\begin{aligned} \min_{y^d} \quad & y^d \cdot v^d \\ \text{such that} \quad & \mathbb{0}^n \leq y^d \cdot M^{d,n} \leq \mathbb{1}^n. \end{aligned} \tag{12}$$

Indeed, the *weak duality theorem* in linear programming ensures that regardless of the feasibility of Eq. (10), it holds that for *every* $y^d$ satisfying the condition of Eq. (12) and *every* $x^n, s^n$ satisfying the conditions of Eq. (11), it is always the case that $y^d \cdot v^d \geq -\mathbb{1}^n \cdot s^n$. So, if *any* $y^d$ can be found satisfying the condition of Eq. (12) such that $y^d \cdot v^d \leq 0$, this serves as a certificate of the infeasibility of Eq. (10).

Now, the matrix $M^{d,n}$, which defines the polytope may exhibit *inherent symmetries*. An inherent symmetry of a matrix is a pair of permutation operations $\pi_{\text{row}}^{d,d}$ and $\pi_{\text{col}}^{n,n}$, acting, respectively, on the row space and column space of the matrix, such that if *both* the row permutation and the column permutation are performed the matrix is invariant. That is,

$$M^{d,n} = \pi_{\text{row}}^{d,d} \cdot M^{d,n} \cdot \pi_{\text{col}}^{n,n}. \tag{13}$$

Whenever such an inherent symmetry can be identified, it can be used to transform feasible solutions of both the primal and dual formulations into new solutions: Suppose we have a collection of vectors $v^d$, $y^d$, $x^n$, $s^n$ such that all of the conditions of both Eq. (11) and Eq. (12) are satisfied. Then, acting on all the vectors with the inherent symmetry leads to a new solution pair to both the primal and dual LP formulations, with the same duality gap (if any). Accordingly, we have that the *symmetrized* inequality $y'^d \geq 0$ where $y'^d := \frac{y^d + \pi_{\text{row}}^{d,d} y^d}{2}$ is also a valid inequality. When $y^d$ is an *optimal* solution to the dual LP in Eq. (12), then symmetrized inequality $y'^d$ is also optimal if $v^d$ is *invariant* under the inherent symmetry operation $\pi_{\text{row}}^{d,d}$.

This is what allows us to restrict the coefficients of the separating hyperplanes. Suppose we find a bunch of different inherent symmetries of the matrix, which defines the polytope; these can be used to construct a *group* with well-defined actions on both the row and column spaces. We can then *twirl* the matrix with respect to this group: We collect columns that map to each other under the group action, and replace each *orbit* of columns with a single new column given by the mean of the orbit. We do the same to the rows. This twirling operation thus yields a substantially smaller matrix, say, $M'^{d,n'}$. Given a vector $v^d$ in the row space of the matrix, we can apply the same twirling to obtain $v'^d$, essentially projecting the vector to the symmetric subspace of the group. We now can obtain a separating hyperplane $y'^d$ by applying the dual formulation of the LP in this symmetric subspace. To convert this hyperplane in the symmetric subspace to a hyperplane in the full row space we de-twirl: namely, each row in a given orbit is uniformly associated with the coefficient of that orbit in the symmetric subspace.

There is no loss of generality whatsoever in using this symmetry-adapted version of the LP if the target vector $v^d$ is also invariant under the group. So, in general, the most efficient way to exploit inherent symmetries in linear programming is to identify the largest symmetry group (acting on both row and column spaces), which leaves both $M^{d,n}$ and $v^d$ invariant.

For more information regarding exploiting symmetry in linear programming see refs. [94-97].

## Robustness to noise added by varying 2-fold coincidence window

We study the behavior of the nonlocality tests over the addition of noise due to the enlargement of the two-fold coincidence window $w_1$. Increasing such a window causes the increase of accidental counts, affecting both the events from the entangled source and those relative to classically correlated signals. From a practical point of view, such noise acts substantially as a white noise on the correlations, that is event pairs, which are uniformly and randomly distributed. Considering such effects, we do expect that at some point, increasing the noise, our witnesses will not be able to detect a nonclassical behavior

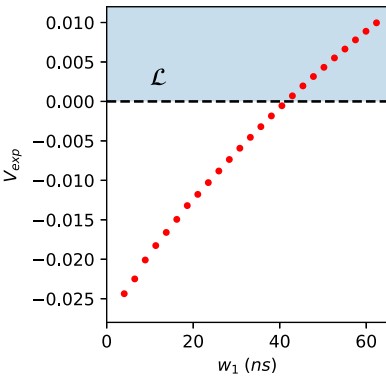

**Fig. 8 | Robustness of the inflation inequality to experimental noise.** In this plot, we show the achieved value for the inequality from inflation technique, as function of the two-fold coincidence window $w_1$. As expected, since this increases uncorrelated 2-fold events, the measured correlations become "local" for large enough windows. The leftmost point corresponds to the result reported in subsection "Bounding measurement dependence and violating an entropic inequality for the triangle network", i.e., a window $w_1 \sim 4.1$ ns, the plotted error bars are calculated through Monte Carlo technique assuming Poissonian statistics and are smaller than the size of the points. The blue shadowed region $\mathcal{L}$ indicates the values obtainable by a classical causal model.

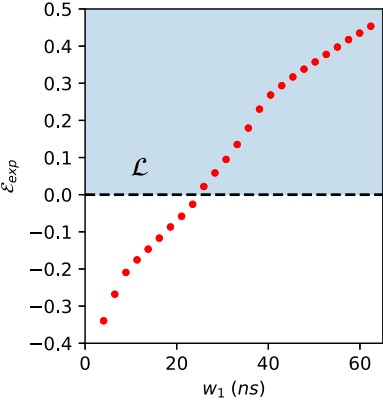

**Fig. 9 | Robustness of the entropic technique to experimental noise.** Values of the quantity $\mathcal{E}_{\exp}$, as defined in Eq. (8), as function of the two-fold coincidence window $w_1$. If $w_1 > 25$ ns, due to the noise, the entropic witness is not able to detect nonclassicality, resulting in a strictly positive value of the quantity $\mathcal{E}_{\exp}$. The leftmost point corresponds to the result reported in subsection "Bounding measurement dependence and violating an entropic inequality for the triangle network", i.e., a window $w_1 \sim 4.1$ ns, the plotted error bars are calculated through Monte Carlo technique assuming Poissonian statistics and are smaller than the size of the points.

anymore. This is, in fact, the case. We show the curve of the violation of the inequality, Eq. (7) and Fig. 5(b) inferred by means of the inflation technique, as a function of the 2-fold window $w_1$ in Fig. 8. The same study is performed with the value of the violation of the entropic inequality in Eq. (8) as shown in Fig. 9.

## Data availability
The data that support the findings of this study are available in the Supplementary Information and from the corresponding author upon request.

## Code availability
All the custom code developed for this study is available from the corresponding author upon request.

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

## Acknowledgements

The authors thank Tamás Kriváchy for discussions about the ML implementation. This work was supported by The John Templeton Foundation via the grant Q-CAUSAL No 61084, via The Quantum Information Structure of Spacetime (QISS) Project (qiss.fr) (the opinions expressed in this publication are those of the author(s) and do not necessarily reflect the views of the John Templeton Foundation) Grant Agreement No. 61466 and via QISS2 Grant Agreement No. 62312, by MIUR via PRIN 2017 (Progetto di Ricerca di Interesse Nazionale): project QUSHIP (2017SRNBRK), by the Regione Lazio programme "Progetti di Gruppi di ricerca" legge Regionale n. 13/2008 (SINFONIA project, prot. n. 85-2017-15200) via LazioInnova spa and by the ERC Advanced Grant QU-BOSS (Grant agreement no. 884676). RC and AC acknowledge the Serrapilheira Institute (Grant No. Serra-1708-15763), the Brazilian National Council for Scientific and Technological Development (CNPq) via the National Institute for Science and Technology on Quantum Information (INCT-IQ) and Grants 307295/2020-6 and No. 311375/2020-0, the Brazilian agencies MCTIC and MEC. Research at Perimeter Institute is supported in part by the Government of Canada through the Department of Innovation, Science and Industry Canada and by the Province of Ontario through the Ministry of Colleges and Universities.

## Author contributions

E.P., D.P., G.R., I.A., G.C., F.S., E.W., R.S., A.C., and R.C. developed the project, E.P., D.P., I.A., A.S., G. Milani, G.C., and F.S. devised the experiment; E.P., D.P., G.R., I.A., A.S., G. Milani, G.C., and F.S. performed the experiment; E.P., D.P., G.R., I.A., A.S., G.C., F.S., E.W., R.S., A.C., G. Moreno, and R.C. performed data-analysis and modeling; E.W. and R.S. developed the theoretical tools of the inflation technique; A.C., G. Moreno, and R.C. developed the theoretical tools of machine learning; all authors discussed the results and contributed to the writing of the manuscript.

## Competing interests

The authors declare no competing interests.
