## [Peer Review File · Nature Communications]

Experimental nonclassicality in a causal network without assuming freedom of choiceREVIEWER COMMENTS

Reviewer #1 (Remarks to the Author):

In the manuscript entitled "Experimental nonclassicality in the triangle causal network", the authors report an experimental violation of local causality in the triangle scenario. In particular, the authors experimentally realize a tripartite distribution and demonstrate its contradiction with the prediction by any local-causal model. For this goal, the authors use an experimental setup with three independent sources for generating entangled photons, and apply two different data-analysis methods. The experimental setup is similar to that used in their previous work (Ref. [43] of the manuscript). The two data-analysis methods are the machine-learning-based test (developed in Ref. [31]) and the inflation technique (developed in Refs. [12,53]). My specific comments are as follows.

First, as to the experimental realization of the tripartite distribution firstly proposed by Fritz, the Fritz distribution reveals a new form of nonclassicality in causal networks, that is, nonclassicality without external inputs. However, in the experimental realization reported in this work, there are external inputs to determine the local measurements on the entangled photons shared between Alice and Bob. To simulate the Fritz distribution, the authors post-select the events where the external inputs coincide with the measurement outcomes at the same stations but on the photons from different sources. In my opinion, this is not a faithful realization of the Fritz distribution. Moreover, if there are external inputs for Alice and Bob to determine their local measurements, it is not necessary to bother a third party Charlie to witness nonclassicality as Alice's and Bob's local measurements with external inputs realize a bipartite distribution which maximally violates the CHSH Bell inequality.

Second, since the data-analysis techniques used in this work were well developed in the previous works (Refs. [12,43,53] of the manuscript), the statement in the abstract "To demonstrate the nonclassicality of our data, we introduce two new techniques ..." is confusing. If I am wrong, please specify in which aspects there are new contributions to these techniques. In addition, if I understand, the machine-learning-based technique may not find the optimal local-causal model with respect to an experimentally observed distribution while the inflation technique guarantees the reliability of the nonclassicality witnessed by experimental data. It is helpful to clarify these points in the manuscript.

Third, I also have several minor comments: i) Can the authors specify the physical distances between any two of Alice, Bob, and Charlie in Figure 2 (b)? Also, where are the three independent sources for entanglement generation located? ii) What is the "single higher pick" as mentioned in the caption of Figure 3? iii) Can the authors justify the statement "When more than three two-fold coincidences are found in the same time window, the additional events are discarded" in Supplementary Note 2?

If the authors can fix the issue detailed in my first comment and clarify their new contributions in data analysis if there are, the work could be appropriate for Nature Communications. Otherwise, I recommend the authors submit this work to a less prestigious journal.

Reviewer #2 (Remarks to the Author):

In this manuscript the authors present an experiment demonstrating nonclassical correlations in a triangle network. The triangle network is a three-party network wherein one assumes causal connections between the three parties such that they are connected by three independent sources in the shape of a triangle. They experimentally achieve this with three independent entangled-pair sources where each source connects two parties. The measurements made within each party are simple separable measurements. It was shown in reference [5] that this situation can lead to non-

classical correlations. This work provides tools to analyze such situations, which go beyond the now standard causal network in a Bell scenario. The work shows how one can use a neural network to model classical correlations in such networks, and derive so-called “causal compatibility inequalities”. I find the work to be relevant and interesting, and I support its publication in Nature Communications. I do have a few points I think the authors should nevertheless address.

1) I'm a little confused about the source independence. The authors state “To justify the assumption of source independence, it is essential to use non-synchronized lasers to pump the generation crystals that act as sources.” They then go on to say that experiments which did not do this require additional device-dependent justifications to claim source independence. However, it seems to me the knowledge that different lasers are used also device dependent knowledge? Furthermore, even if I had photons generated using different pump lasers, I could imagine some non-local interaction which later generates correlations between the photons. I think ideally one would devise some set of measurements to verify this condition.

2) I think the authors should include a few more experimental details in the main body, or the methods section. The use of the three uncorrelated sources using both CW and pulsed lasers is quite unusual. I did not see any remarks on the 6-fold rate or the total acquisition time. The only remark related to this is total counts acquired which is mentioned in the supplement. These are very important as the total number of counts acquired is used to calculate their error bars. Furthermore, the 63us window is quite large and, again, unusual. The authors describe their choice of the window well in the supplement, but a small remark about its significance should be made in the main body. They could also comment on how this choice affects their 6-fold rate (even in the supplement).

3) At the start of the “Fritz Distribution measurement” section, the authors say “Based on the result of this measurement, one of the two Bell observables ... are measured.” This confused me, as it seems to imply some sort of feed-forward, and earlier in the paper the authors say “Here, in stark contrast to the Bell scenario, the parties implement a single measurement, rather than implementing one of a set of incompatible measurements.” Can the authors clarify this point?

Reviewer #3 (Remarks to the Author):

A causal network consists of sources emitting signals, and parties performing measurements on the received signals. The question of characterizing correlations among measurement outcomes of the parties that are compatible with a given causal network is of fundamental interest, particularly because of the phenomenon of network nonlocality.

In the paper under review, the authors consider the triangle network and experimentally demonstrate that this network exhibits nonlocality. The triangle network consists of three parties any pair of which share a source. Fritz's distribution is a particular distribution on the outcomes of the three parties that is rooted in the CHSH correlation, and is a nonlocal distribution. The authors in this paper, perform an experiment to observe Fritz's distribution in the triangle network using sources emitting entangled states. The resulting correlation is of course a noisy version of Fritz's distribution. Then the authors give evidences/proofs to show that even this noisy distribution is nonlocal.

The authors first use machine learning techniques to find optimal response functions in a hypothetical classical scenario that may realize the noisy Fritz's distribution. They conclude that with these methods one cannot get close to the desired distribution. This method, although heuristic gives an evidence that the outcome distribution of the experiment is nonlocal.

The next approach of the authors is via the inflation technique. Inflation is a method for deriving a hierarchy of necessary conditions on local distributions, which interesting are sufficient too. The authors use a second order inflation which given the outcome distribution of the experiment, is a linear program. The authors observe that this linear program is infeasible and conclude that the distribution is nonlocal. To prove infeasibility they use Farkas lemma, and explicitly find an inequality

that holds for all local correlations but is violated by the distribution under study. I found this part of the paper particularly interesting.

I have a major concern about this paper. In the experimental implementation of Fritz's distribution, the authors take all the shared sources to be entangled singlet states. However, in principle two of the sources can be assumed to be classical. While classical sources can be simulated by quantum ones, this would introduce extra noise. By replacing these two sources with classical ones, the associated measurement outcomes would have perfect anti-correlation, resulting in much less noise in the outcome distribution. In fact, it seems to me that the authors used a complicated experimental setup for something that could be done much easier and with a reduced noise. Having said that, even if the author would've replaced two of the quantum sources with classical ones, I'm not sure if that experiment would've been interesting since such an experiment would be essentially a Bell experiment.

I have three other main comments:

As far as I understand, an important parameter in the neural networks considered in the paper is the size of inputs; the alphabet sizes of hidden variables Λ_{AB} , Λ_{BC} and Λ_{AC} . Maybe I'm missing something, but I don't see any comment on this in the paper.

In the inflation part, the dual vector obtained via Farkas lemma is quite interesting since it is very symmetric. I wished the authors would've commented more on the features of this vector and the resulting inequality. In particular, I liked to see an analytic proof of that inequality. My point is that the nonlocality of Fritz's distribution is rooted in the CHSH inequality. Moreover, as far as I understand, the CHSH inequality itself can be proven using a second order inflation. Thus it is not a surprise that the locality of a noisy version of Fritz's distribution can be refuted using a second order inflation. But since the inequality obtained via this method is quite symmetric (and the CHSH inequality has an analytic proof), it would be interesting to find a direct analytic proof of the new inequality too. Such a proof may advance our understating of network nonlocality.

It also would be interesting if the authors would've compared the results of the two methods. The machine learning technique gives a (heuristic) bound on the visibility of the distribution under study. The inequality found via the inflation technique also gives such a bound. Now my question is how these two bounds compare to each other?

And here are some minor comments:

Line 66: what "new set of data analysis techniques" refer to? If they are the machine learning and inflation techniques, both of them appear in previous works.

Line 99: "is demonstrate" → to demonstrate

Line 117: "answers the question" → and answers

Line 185: "Positive-valued-measure" → positive operator-valued measure

Line 206: shouldn't a_0 in " $p(a_1, b_1 | a_0, c_0)$ " be c_1 ?

First paragraph of Supplementary Note 4: "an ansatz causal structures"

Supplementary Note 6 is very brief and is not clear to me. Expanding this part and e.g. giving the proof of equation (14) may make it clearer.

Reply to Reviewer 1

In the manuscript entitled “Experimental nonclassicality in the triangle causal network”, the authors report an experimental violation of local causality in the triangle scenario. In particular, the authors experimentally realize a tripartite distribution and demonstrate its contradiction with the prediction by any local-causal model. For this goal, the authors use an experimental setup with three independent sources for generating entangled photons, and apply two different data-analysis methods. The experimental setup is similar to that used in their previous work (Ref. [43] of the manuscript). The two data-analysis methods are the machine-learning-based test (developed in Ref. [31]) and the inflation technique (developed in Refs. [12,53]). My specific comments are as follows.

First, as to the experimental realization of the tripartite distribution firstly proposed by Fritz, the Fritz distribution reveals a new form of nonclassicality in causal networks, that is, nonclassicality without external inputs. However, in the experimental realization reported in this work, there are external inputs to determine the local measurements on the entangled photons shared between Alice and Bob. To simulate the Fritz distribution, the authors post-select the events where the external inputs coincide with the measurement outcomes at the same stations but on the photons from different sources. In my opinion, this is not a faithful realization of the Fritz distribution. Moreover, if there are external inputs for Alice and Bob to determine their local measurements, it is not necessary to bother a third party Charlie to witness nonclassicality as Alice’s and Bob’s local measurements with external inputs realize a bipartite distribution which maximally violates the CHSH Bell inequality.

We thank the Reviewer for the detailed report and for the pertinent questions and comments that certainly have helped us to improve and clarify the main goal of our manuscript.

We agree that the post-selection opens a loophole in our demonstration in the same way that low detection efficiency does for standard Bell tests. We certainly do not achieve a loophole free demonstration of non-classicality in the triangle network but neither was that our goal. Rather, it was to implement the quantum triangle network, use it to generate quantum correlations without a classical explanation (up to loopholes) and most importantly develop tools to analyze this experimental data and prove its non-classicality.

In this sense we highlight that our goal was not to provide a faithful realization of the Fritz distribution. It serves only as a guide to what experiment could be done to generate non-classical correlations in our triangle network. In particular, notice that the argument provided by Fritz to prove the nonclassicality of his distribution requires perfect correlation between Charlie’s output c_0 and Alice’s output a_0 as well as perfect correlation between Charlie’s output c_1 and Bob’s output b_0 . Only under this strict constraint is the Fritz distribution known to be nonclassical. If this constraint could be satisfied, we agree that simply testing the CHSH inequality between Alice and Bob would be enough to witness nonclassicality and that Charlie’s role would be superfluous. However, in any experimental implementation, such perfect correlation is always out of reach. In the new version of the paper we clarified the above points at the end of the Introduction and section II-C.

A similar argument could be given, for instance, for Bell’s original proof of his theorem or for the GHZ paradox. Both require perfect correlations to witness non-classicality and thus serve only as gedanken experiments. It was only with the discovery of the CHSH and Mermin inequalities that these theoretical arguments could be extended to demonstrate nonclassicality in the laboratory. Within this context, one can argue that our analysis achieves the analogous extension for the Fritz argument.

As briefly noted in Supplementary Note 6, the perfect correlation in Fritz’s proof is required to map the triangle network to the usual Bell network. The absence of perfect correlations can be formally mapped to a Bell scenario with measurement dependence (also known as “lack of free-will”). More precisely, we show that the triangle network can be employed to provide lower and upper tight bounds to the level of measurement dependence, as

$$I(a_0, b_0) \leq I(a_0, b_0 : \lambda_{ab}) \leq H(a_0, b_0 | c_0, c_1), \quad (1)$$

where $I(a_0, b_0 : \lambda_{ab})$ is the mutual information between Alice and Bob inputs and the source λ_{ab} and is a measure of the level of measurement dependence if we interpret our experiment via this mapping to a Bell scenario. From this bound, we can estimate the level of measurement dependence in our experiment to be $0.0034 \pm 0.0061 \leq I(a_0, b_0 : \lambda_{ab}) \leq 0.206 \pm 0.001$.

Given this, one could argue then that our experiment is a new kind of Bell test where one explicitly allows for a relaxation of the measurement independence assumption in Bell’s theorem. That is, we would be proving a stronger form of non-classicality as compared with the standard Bell notion and as a consequence also a different form to that of

the Fritz distribution. Notice, however, that it is known that even a very small amount of measurement dependence is already enough to simulate even the maximal violation of a Bell inequality. For instance, in [1], it has been shown that 0.046 bits of correlation are enough to achieve the maximum violation of the CHSH inequality. In our experiment the measurement dependence is between $0.0034 \pm 0.0061 \leq I(a_0, b_0 : \lambda_{ab}) \leq 0.206 \pm 0.001$. Thus, in the device-independent perspective (based on the experimental data alone) we cannot exclude the possibility that the violation of the CHSH inequality is due to a classical mechanism rather than a non-classical behaviour. In short, the simple violation of the CHSH inequality is not enough to witness non-classicality.

Altogether, the fact that we do not implement the exact Fritz distribution is a feature rather than a failure of our implementation. If the outcomes of Alice and Bob are perfectly correlated with that of Charlie, then the non-classicality of the Fritz distribution can indeed be witnessed by violating the CHSH inequality. But that is not the case in our experiment. The absence of perfect correlations can be understood as measurement dependence implying that the CHSH inequality can no longer be seen as a valid non-classicality witness. We believe that this is a strong argument showing that our experiment is witnessing a new form of non-classical behaviour and that the new analysis tools we employ (machine learning and the inflation technique) are really required.

In the first version of the manuscript this discussion appears only in the Supplementary material. But in view of the questions and comments made by the Reviewer we have added a few sentences explaining that in more details and also include a reference ([44]) to the posted theoretical paper with the complete and detailed analysis of it (which is much beyond the scope of the current paper).

In particular, we have added the following text: “Any experiment that aims to realize the Fritz distribution in the triangle scenario aims to realize the ideal states and measurements specified above, but due to the inevitability of noise, the states and measurements that are actually implemented are necessarily noisy versions of these. This implies that the correlations between a_0 and c_0 and between b_0 and c_1 will not be perfect, which in turn blocks the inference from the classical realizability of $p(a, b, c)$ in the triangle scenario to the classical realizability of $p(a_1, b_1 | a_0, b_0)$ in the standard Bell scenario. As such, to witness nonclassicality in such an experiment, one must go beyond the techniques that witness nonclassicality in a standard Bell experiment. Note, however, that there is an opportunity for using data-analysis techniques for a nonstandard type of Bell experiment, namely, one wherein the failure of measurement independence can be bounded [43]. An example of such an analysis is presented in Ref. [44]. As explained in more detail in the Supplementary Information, however, the techniques developed in Ref. [44] lack the sensitivity to be able to witness the nonclassicality of our experimental data. In order to be able to witness the nonclassicality of our data, therefore, we cannot rely on standard Bell inequalities nor even the inequalities that arise in a nonstandard Bell experiment wherein the failure of measurement independence can be bounded [44]. This is why we must have recourse to new data-analysis techniques, such as those presented in Secs. IV A and IV B.”

Second, since the data-analysis techniques used in this work were well developed in the previous works (Refs. [12,43,53] of the manuscript), the statement in the abstract “To demonstrate the non-classicality of our data, we introduce two new techniques ...” is confusing. If I am wrong, please specify in which aspects there are new contributions to these techniques. In addition, if I understand, the machine-learning-based technique may not find the optimal local-causal model with respect to an experimentally observed distribution while the inflation technique guarantees the reliability of the nonclassicality witnessed by experimental data. It is helpful to clarify these points in the manuscript.

We agree with the Referee that these data-analysis techniques have been introduced before in a purely theoretical context. However, in order to use them (for the first time) to analyze experimental data we had to adapt and improve them in a number of ways described in details below. We also agree that the sentence in the abstract may be confusing and for this reason we have changed it to this: “To demonstrate the non-classicality of our data (modulo the detector loophole), we adapt and improve two known techniques...”

For the machine learning-based witness, we improved the performances of the approach presented in [2] by exploiting an ensemble of neural networks with distinct number of layers and neurons. The advantage of using an assemblage of oracles is described in the Supplementary Note 4, where it is shown that the best kind of architecture yielding the minimum MSE distance changes for distributions with different visibilities (See Supplementary Table I). The original approach in [2] works well for perfect and symmetric probability distributions but performs poorly in our real experimental data. In fact, more advanced neural networks architectures could be used as well, such as convolutional neural networks (CNNs), recurrent neural networks (RNNs) and so on, but this would require a dedicated investigation that would be far beyond the scope of this paper, although the implementation of an ensemble of MLPs is already a natural advance of the inaugural method shown in Ref. [2].

In parallel, the causal compatibility inequalities we have derived are data-seeded, and this changes the usual paradigm.

We started from the causal structure and some experimental data, and then, thanks to the Inflation technique, we constructed an inequality, valid for classical models in the triangle scenario and able to be violated by the measured data. In a typical experiment, it would be the other way around: first derive an inequality and then try to improve the experimental setting in order to violate it. For instance, in [3] the inflation technique has been used to derive an inequality violated by the (perfect) Fritz distribution but it would be of little use in our case, since it required entangled states with almost perfect visibilities.

The Referee is right about the machine learning approach. It is a numerical tool and as such cannot be considered as a stand-alone proof of non-classicality. In turn, the violation of the data-seeded causal inequality is an unambiguous proof of the non-classical nature of our experiment. We clarified this point in the final part of the section IV-A.

Since the employed approaches are likely to be a powerful tool for demonstrating the non-classicality in general quantum networks, and as this is the first time that they are employed in a novel causal structure with real experimental data, we think that our contribution deserves to be considered as an important progress in the field.

Third, I also have several minor comments:

i) Can the authors specify the physical distances between any two of Alice, Bob, and Charlie in Figure 2 (b)? Also, where are the three independent sources for entanglement generation located?

The physical distances between the sources are the following: $\rho_{AC} - \rho_{BC} \sim 4m$, $\rho_{AB} - \rho_{AC} \sim 13m$, $\rho_{AB} - \rho_{BC} \sim 9m$. Each source is located on different optical tables and source ρ_{AB} lies within a different laboratory with respect to the one containing ρ_{AC} and ρ_{BC} . The distances between the measurement stations are the following: Alice-Bob $\sim 10m$, Bob-Charlie $\sim 3m$ and Alice-Charlie $\sim 12m$.

In order to clarify this point, we have included such information in the Supplementary Note 1.

ii) What is the “single higher pick” as mentioned in the caption of Figure 3?

We thank the Referee for pointing out this typo: it refers to the peak attributed to the probability $p(a = 3, b = 2)$. We corrected the typo and explicitly indicated the probability term.

iii) Can the authors justify the statement “When more than three two-fold coincidences are found in the same time window, the additional events are discarded” in Supplementary Note 2?

When we look for 6-fold events, within a given time interval, we aim at detecting three 2-fold events, each one belonging to a different source. However, within such time window, more than one 2-fold coincidence can be generated by the same source, due to the very different rates characterizing them. If this happens, we select only the coincidence which is detected first. Let us note that this constitutes a simple and useful convention to avoid ambiguities in our definition of 6-fold events and it is an unbiased choice.

In order to clarify this aspect, we added the following paragraph in the Supplementary Note 2:

“More specifically, when more than one coincidence from the same source are found within the window interval, we select only the first detected. Selecting the first coincidence represents an arbitrary and unbiased choice, that is a useful convention for resolving possible ambiguities in the definition of six-fold events.”

If the authors can fix the issue detailed in my first comment and clarify their new contributions in data analysis if there are, the work could be appropriate for Nature Communications. Otherwise, I recommend the authors submit this work to a less prestigious journal.

We thank the Reviewer for the relevant questions and comments. They have helped us to clarify both the motivation for our experimental setup and the new contributions of the data analysis. We hope that with these clarifications and improvement the Reviewer can judge our work suitable for publication in Nature Communications.

Reply to Reviewer 2

In this manuscript the authors present an experiment demonstrating nonclassical correlations in a triangle network. The triangle network is a three-party network wherein one assumes causal connections between the three parties such that they are connected by three independent sources in the shape of a triangle. They experimentally achieve this with three independent entangled-pair sources where each source connects two parties. The measurements made within each party are simple separable measurements. It was shown in reference [5] that this situation can lead to non-classical correlations. This work provides tools to analyze such situations, which go beyond the now standard causal network in a Bell scenario. The work shows how one can use a neural network to model classical correlations in such networks, and derive so-called “causal compatibility inequalities”. I find the work to be relevant and interesting, and I support its publication in Nature Communications. I do have a few points I think the authors should nevertheless address.

We thank the Reviewer for the relevant questions and comments as well as for finding this work relevant and interesting, hence suggesting its publication in Nature Communications.

1) I’m a little confused about the source independence. The authors state “To justify the assumption of source independence, it is essential to use non-synchronized lasers to pump the generation crystals that act as sources.” They then go on to say that experiments which did not do this require additional device-dependent justifications to claim source independence. However, it seems to me the knowledge that different lasers are used also device dependent knowledge? Furthermore, even if I had photons generated using different pump lasers, I could imagine some non-local interaction which later generates correlations between the photons. I think ideally one would devise some set of measurements to verify this condition.

We thank the Reviewer for this relevant question. In a device-independent setting, properties can be inferred from observed statistics, relying only on causal assumptions. In the experimental implementation of a device-independent protocol, what the experimenter can do is to enforce such causal assumptions.

We notice that a similar assumption about source independence also appears in a standard Bell experiment. The two causal assumptions in Bell’s theorem are locality and measurement independence (“free-will”). The first can be enforced by a physical principle, such as relativity in space-like separated measurements. The independence of the sources generating the inputs for Alice and Bob and the source generating the entangled state to be measured, however, cannot be enforced by physical principles, as a super deterministic model, where all of the sources are affected by a hidden variable, does not provide any testable mathematical constraint and, therefore, is not falsifiable through data. However, we can still reach different levels of plausibility of this independence assumption. For example, if one considers a scenario in which the sources are pumped by the same laser, the assumption of independence among the sources is less justifiable, with respect to our implementation, where different and physically separated devices are employed and one could in principle reach a space-like separation of the states generation. If one were to use the same laser, then in order to justify the independence, one would need to rely on the knowledge of the inner working of the sources; this is the sense in which we use the term “device-dependent justification” (a term that we changed in order to avoid confusion among the readers)

In order to clarify this aspect, we modified the paragraph in the following way:

“This is the reason why the first experimental realization of quantum networks [4–6] actually involved a single laser source, thereby requiring a justification for the supposed independence of the generated quantum states that relies on the knowledge of the inner process of photon generation. Hence, using different and spatially separated pump lasers allows to strongly enforce the independence of the sources, also having direct applications in quantum communication protocols”.

2) I think the authors should include a few more experimental details in the main body, or the methods section. The use of the three uncorrelated sources using both CW and pulsed lasers is quite unusual. I did not see any remarks on the 6-fold rate or the total acquisition time. The only remark related to this is total counts acquired which is mentioned in the supplement. These are very important as the total number of counts acquired is used to calculate their error bars. Furthermore, the 63us window is quite large and, again, unusual. The authors describe their choice of the window well in the supplement, but a small remark about its significance should be made in the main body. They could also comment on how this choice affects their 6-fold rate (even in the supplement).

We followed the Reviewer's suggestion and included more experimental details in the main text and methods sections:

- CW and pulsed sources: A strength of our demonstration is that it relies only on separable measurements. In this way, it is scalable and paves the way for the realization of larger and more complex networks. In particular, the employed entangled photons sources can in principle be completely different in terms of generation rates, pulsed or continuous wave pumps and generated wavelengths. For this reason we employed sources with both CW and pulsed lasers. We specify it in the section III of the main text by adding the following paragraph:

“Our demonstration relies only on separable measurements on the pair of photons within a given lab, and this allows the scalability and reliability of the approach for real quantum networks. In particular, the employed entangled photons sources can in principle be completely different in terms of generation rates, pulsed or continuous wave (cw) pumps and generated wavelengths. For this reason we employed sources with both cw and pulsed lasers.”

- 6-fold rate and acquisition time: The 6-fold rate with a window equal to $63.6\mu\text{s}$ is 31.6Hz and total measurement time is 25587s , corresponding to a total number of events $\sim 8 \cdot 10^5$. We added this information in the first paragraph of the section “Results” of the main text.

- 6-fold coincidence window: The value of the chosen window, $63.6\mu\text{s}$, represents a good compromise between the detected rate and the closeness in time of the measured events. As the coincidence window increases (up to a critical value), the 6-fold events are likely to be found within it and, consequently the statistics improves. On the other side, as we reduces the coincidence window, the events are closer to space-like separation condition. In order to make this clear, we added the following paragraph in the section “Results”:

“Such a choice of value of the 6-fold coincidence window represents a compromise between the requirements that the events generated by the three sources were as close as possible in time to approximate the simultaneity, i.e. space-like separation, of the measurements (narrow windows), and, on the other side, the necessity to detect a sufficient number of events in order to have sufficiently small errors on the measured probabilities (larger windows).”

Regarding the rate, we studied windows from $2.835\mu\text{s}$ to $63.585\mu\text{s}$ with a corresponding detected 6-fold coincidence rate of 3.59Hz and 31.62Hz , respectively. We added this information in the Supplementary Note 2 where we insert the following:

“The plot clearly shows that a significant violation can be obtained for all the considered values of the window, ranging from $2.835\mu\text{s}$ to $63.585\mu\text{s}$ with a corresponding detected 6-fold coincidence rate of 3.59Hz and 31.62Hz , respectively.”

3) At the start of the “Fritz Distribution measurement” section, the authors say “Based on the result of this measurement, one of the two Bell observables ... are measured.” This confused me, as it seems to imply some sort of feed-forward, and earlier in the paper the authors say “Here, in stark contrast to the Bell scenario, the parties implement a single measurement, rather than implementing one of a set of incompatible measurements.” Can the authors clarify this point?

One possible way to implement the Fritz distribution is by means of a feed-forward mechanism. Alice first measures her subsystem shared with Charlie and then, depending on the outcome, changes the measurement applied to her subsystem shared with Bob. Similarly does Bob, by measuring photons shared with Charlie and Alice. The sentence in the Introduction “Here, in stark contrast to the Bell scenario, the parties implement a single measurement, rather than implementing one of a set of incompatible measurements.” is referred to the fact that, unlike Bell scenario, Fritz distribution does not need external inputs to be realized. The feed-forward is driven by the results of (fixed) measurements on subsystems and no free external inputs have to be provided by Alice, Bob and Charlie. As we now discuss more clearly in the text, this feed-forward measurement can be seen as a single fixed POVM (that is, without any external inputs).

FIG. 1: Number of 6-fold coincidence as a function of the window: Rate in Hz of the 6-fold coincidence events as function of the time window, where three coincidences from the three sources are considered simultaneous.

Reply to Reviewer 3

A causal network consists of sources emitting signals, and parties performing measurements on the received signals. The question of characterizing correlations among measurement outcomes of the parties that are compatible with a given causal network is of fundamental interest, particularly because of the phenomenon of network nonlocality. In the paper under review, the authors consider the triangle network and experimentally demonstrate that this network exhibits nonlocality. The triangle network consists of three parties any pair of which share a source. Fritz's distribution is a particular distribution on the outcomes of the three parties that is rooted in the CHSH correlation, and is a nonlocal distribution. The authors in this paper, perform an experiment to observe Fritz's distribution in the triangle network using sources emitting entangled states. The resulting correlation is of course a noisy version of Fritz's distribution. Then the authors give evidences/proofs to show that even this noisy distribution is nonlocal. By violating the causal compatibility inequalities we demonstrate that it is not like a simple Bell scenario: such inequalities refer exactly to the triangle scenario. The authors first use machine learning techniques to find optimal response functions in a hypothetical classical scenario that may realize the noisy Fritz's distribution. They conclude that with these methods one cannot get close to the desired distribution. This method, although heuristic gives an evidence that the outcome distribution of the experiment is non-local. The next approach of the authors is via the inflation technique. Inflation is a method for deriving a hierarchy of necessary conditions on local distributions, which interesting are sufficient too. The authors use a second order inflation which given the outcome distribution of the experiment, is a linear program. The authors observe that this linear program is infeasible and conclude that the distribution is non-local. To prove infeasibility they use Farkas' lemma, and explicitly find an inequality that holds for all local correlations but is violated by the distribution under study. I found this part of the paper particularly interesting.

We thank the Referee for acknowledging the interest of the techniques employed to demonstrate the non-classicality of the realize triangle network. In the following they can find the answers to all the raised points.

I have a major concern about this paper. In the experimental implementation of Fritz's distribution, the authors take all the shared sources to be entangled singlet states. However, in principle two of the sources can be assumed to be classical. While classical sources can be simulated by quantum ones, this would introduce extra noise. By replacing these two sources with classical ones, the associated measurement outcomes would have perfect anti-correlation, resulting in much less noise in the outcome

distribution. In fact, it seems to me that the authors used a complicated experimental setup for something that could be done much easier and with a reduced noise. Having said that, even if the author would've replaced two of the quantum sources with classical ones, I'm not sure if that experiment would've been interesting since such an experiment would be essentially a Bell experiment.

We thank the Reviewer for giving us the chance to clarify this important experimental aspect. A related aspect of this question has been done by the Reviewer 1 and we ask Reviewer 3 to also check our reply there.

We agree with them that, regarding the realization of Fritz distribution, the two quantum sources can be replaced by two classical random number generators. Our demonstration indeed does not rely on the entanglement of two of the sources. In the case of classical sources, as the Reviewer correctly states, the correlations would be almost perfect (for instance using an electronic signal). However, there are two crucial points that have to be considered and that motivates the employment of entangled sources.

First, the fact that we do not implement the exact Fritz distribution is a feat rather than a failure of our implementation. If the outcomes of Alice and Bob are perfectly correlated with that of Charlie, then the non-classicality of the Fritz distribution can indeed be witnessed by violating the CHSH inequality. And for that, as the Reviewer correctly says, it would be better to use classical sources with less noise. But that is not the case in our experiment. The absence of perfect correlations can be understood as measurement dependence implying that the CHSH inequality can no longer be seen as valid non-classicality witness (please refer to our more detailed answer to Reviewer 1). The fact we have noisy data implies we cannot rely on a standard Bell test.

The second crucial point is that both our experimental setup as well as the data analysis tools are not limited to the Fritz distribution. Here, the scope is to develop and exploit techniques able to witness nonclassicality in experimental noisy data. The realized noisy Fritz distribution is only a means to demonstrate such nonclassicality in a real setup that is able to share three entangled states and can in principle be useful also for realizing other quantum distributions. Hence, our analysis, having non-perfect correlations, cannot rely on Fritz' proof of nonclassicality and goes beyond the simple realization of Fritz distribution with perfect correlations. In this way, we demonstrate general analysis tools for a real and versatile quantum network, where each involved source is able to generate and distribute quantum states.

In the final part of Section III, we have added:

*“Note that in order to realize the Fritz distribution, it is sufficient to share entanglement only between Alice and Bob's labs, since Alice and Charlie as well as Bob and Charlie can merely share classical correlations. Moreover, using such classical sources, e.g., a doubled electronic signal, makes it possible to achieve correlations between Alice and Charlie and between Bob and Charlie that can be perfect for the duration of the experiment (recall that perfect correlation is required for the logic of Fritz's argument to go through). For such a realization, demonstrating non-classicality in the triangle network could be achieved in the manner described by Fritz, and thus would boil down to violating a standard Bell inequality. We did not use this approach here, as it is the goal of our work to introduce and validate data analysis techniques that are applicable for *any* example of a quantum-classical gap in the triangle scenario, including gaps based on distributions that, unlike Fritz's, *do* require entanglement in all three sources.”*

I have three other main comments:

1) As far as I understand, an important parameter in the neural networks considered in the paper is the size of inputs; the alphabet sizes of hidden variables Λ_{AB} , Λ_{BC} and Λ_{AC} . Maybe I'm missing something, but I don't see any comment on this in the paper.

We thank the Referee for pointing it out. In fact, we omitted this important discussion in the main text, presenting it only in the Supplementary Note 4. In this new version of the paper, we bring the discussion to the main text by adding the following sentence:

“From the machine learning perspective, the input layers to the multilayer perceptrons (MLPs) are composed of the independent uniformly distributed random numbers in the unit interval, i.e. $\lambda_{AB}, \lambda_{BC}, \lambda_{AC} \in [0, 1]$, with the restriction in the flow of information imposed by the triangle network: each source only sends information to two parties of the three. In this manner, Alice receives $(\lambda_{AB}, \lambda_{AC})$, Bob receives $(\lambda_{AB}, \lambda_{BC})$ and Charlie receives $(\lambda_{BC}, \lambda_{AC})$. Therefore, individual inputs $\in \mathbb{R}^2$ (they have length 2). For training, we provide batches of $(N_{\text{batch}}, 2)$ dimension for the corresponding MLP for each party.”

2) In the inflation part, the dual vector obtained via Farkas' lemma is quite interesting since it is very symmetric. I wished the authors would've commented more on the features of this vector and the resulting inequality. In particular, I liked to see an analytic proof of that inequality. My point is that the nonlocality of Fritz's distribution is rooted in the CHSH inequality. Moreover, as far as I understand, the CHSH inequality itself can be proven using a second order inflation. Thus it is not a surprise that the locality of a noisy version of Fritz's distribution can be refuted using a second order inflation. But since the inequality obtained via this method is quite symmetric (and the CHSH

inequality has an analytic proof), it would be interesting to find a direct analytic proof of the new inequality too. Such a proof may advance our understating of network non-locality.

We share the referee’s intuition that further insight into qualitative features of this inequality would be useful for advancing our understanding of network non-locality. Attaining a qualitative understanding — such as an analytic proof of the inequality without automated verification through linear programming — is an aspirational desideratum which we defer to future research. This manuscript is meant to showcase a technological demonstration that experimental network non-locality can be confidently witnessed; we certainly endorse parallel research efforts to improve our foundational understanding. We also refer the referee to Ref. [3, Sec. VII-A], where an inflation-derived inequality witnessing the *perfect* Fritz distribution *is* more apparently rooted in the CHSH inequality.

3) It also would be interesting if the authors would’ve compared the results of the two methods. The machine learning technique gives a (heuristic) bound on the visibility of the distribution under study. The inequality found via the inflation technique also gives such a bound. Now my question is how these two bounds compare to each other?

Indeed the two methods can be compared. But please notice that the machine learning prediction can only give a rough estimate because the distance will never be identically zero (even if the distribution is classical and thus can in principle be reproduced exactly by the neural network, it still returns an mean squared error in the order of 10^{-9}). For the perfect Fritz distribution the analytical values for the critical visibility is known to be $v_{crit} \approx 0.7071$, the same result approximately obtained by the machine learning approach. For the actual experimental distribution one can see that the critical visibility is around the same point.

Using the 2-copy instance of the inflation technique, the non-classicality of the perfect Fritz distribution is only witnessed down to critical visibility 0.862, as noted in the Supplementary Materials. This is weaker than the machine learning approach, as would be expected, since finite-level inflation is a relaxation of classical causal compatibility. We have also computed the visibility of the experimental distribution against mixing with white noise using the experiment-optimized inequality which appears in the Supplementary Materials. We find a critical visibility of $v_{crit} = 0.930$. Again, this weaker visibility relative to the machine learning approach is the anticipated price paid by inflation in order to guarantee incompatibility without numerical heuristics. We have elected to omit this numerical details in the revised manuscript, as we do not feel that it offers insight beyond the current Supplementary Materials discussion on the critical visibility of the perfect Fritz distribution.

And here are some minor comments:

Line 66: what “new set of data analysis techniques” refer to? If they are the machine learning and inflation techniques, both of them appear in previous works.

We thank the Reviewer for raising this point that was also inquired by Reviewer 1. We agree with the Referee that these data-analysis techniques have been introduced before in a purely theoretical context. However, in order to use them (for the first time) to analyze experimental data we had to adapt and improve them in a number of ways described in details below. We also agree that the sentence in abstract can be confusing and for this reason we have changed it for “To demonstrate the non-classicality of our data (modulo the detector loophole), we adapt and improve two known techniques...”

For the machine learning-based witness, we improved the performances of the approach presented in [2] by exploiting an ensemble of neural networks with distinct number of layers and neurons. The advantage of using an assemblage of oracles is described in the Supplementary Note 4, where it is shown that the best kind of architecture yielding the minimum MSE distance changes for distributions with different visibilities (See Supplementary Table I). The original approach in [2] works well for perfect and symmetric probability distribution but performs worse in our real data, because a fix neural network architecture may not be enough. In fact, more advanced neural networks architectures could be used as well, such as convolutional neural networks (CNNs), recurrent neural networks (RNNs) and so on, but this would require a dedicated investigation that would be far beyond the scope of this paper, although the implementation of an ensemble of MLPs is already a natural advance of the inaugural method shown in Ref. [2].

In parallel, the causal compatibility inequalities we have derived are data-seeded, and this changes the usual paradigm. We started from the causal structure and some experimental data, and then, thanks to the Inflation technique, we constructed an inequality, valid for classical models in the triangle scenario and able to be violated by the measured data. In a typical experiment we would have the other way around, first derive an inequality and then try to improve the experimental setting in order to violated. For instance, in [3] the inflation technique has been used to derive an inequality violated by the (perfect) Fritz distribution but it would be of little use in our case, since it required entangled states with almost perfect visibilities.

Line 99: “is demonstrate” → to demonstrate. **Line 117:** “answers the question” → and answers. **Line**

185: “Positive-valued-measure” → positive operator-valued measure

We thank the Reviewer for noticing these typos, which we fixed.

Line 206: shouldn't a_0 in $p(a_1, b_1|a_0, c_0)$ be c_1 ?

We corrected the typo changing “ $p(a_1, b_1|a_0, c_0)$ ” with “ $p(a_1, b_1|a_0, b_0)$ ”, since for the Fritz distribution, the choice of measurements in Alice and Bob stations are given by a_0 and b_0 , respectively.

First paragraph of Supplementary Note 4: “an ansatz causal structures”

We fixed the reported typo.

Supplementary Note 6 is very brief and is not clear to me. Expanding this part and e.g. giving the proof of equation (14) may make it clearer.

As mentioned in our reply above, we have decided to substantially expand this supplementary Note 6 and write a theoretical manuscript (recently posted on arxiv and cited as Ref [44] in the new version of the manuscript) proving these bounds along many other results. In any case, we have followed the Reviewer's suggestion and added a proof of equation (14) along with more details. We also have now a more detailed discussion of its role in the main manuscript.

List of Changes:

- We have modified the introduction in order to highlight the goal and novelty of our work.
- We modified the main text in order to take into account the considerations by Referees. In particular, we stress which is the purpose of our work, its relation with Fritz distribution and the difference with respect to a standard Bell test. Given the purposes, we carefully explain why we performed the experiment as it has been done.
- We have included technical details within the Supplementary Information adding the additional informations required by Referees.
- We have included details on the experimental setup as requested by the reviewers.
- We have added in the main text a citation to [44], an article recently posted on the arxiv, which shows that testing the nonclassicality of a Fritz-type distribution that does not achieve perfect correlations between the setting variable in Alice's (Bob's) lab and Charlie's lab can be related to a Bell-type scenario but only a nonstandard one wherein there is a failure of measurement dependence. Our discussion of these ideas should clarify the differences between our experiment and a standard Bell scenario. This new article also demonstrates the relevance and richness of perspectives of our work.
- We have corrected all the found typos.

-
- [1] M. J. W. Hall, “Relaxed bell inequalities and kochen-specker theorems,” *Phys. Rev. A*, vol. 84, p. 022102, Aug 2011.
- [2] T. Kriváchy, Y. Cai, D. Cavalcanti, A. Tavakoli, N. Gisin, and N. Brunner, “A neural network oracle for quantum nonlocality problems in networks,” *npj Quantum Information*, vol. 6, p. 70, Aug 2020.
- [3] T. C. Fraser and E. Wolfe, “Causal compatibility inequalities admitting quantum violations in the triangle structure,” *Phys. Rev. A*, vol. 98, no. 2, p. 022113, 2018.
- [4] G. Carvacho, F. Graffitti, V. D'Ambrosio, and F. Hiesmayr, B. Sciarrino, “Experimental investigation on the geometry of GHZ states,” *Scientific Reports*, vol. 7, p. 13265, 2017.
- [5] D. J. Saunders, A. J. Bennet, C. Branciard, and G. J. Pryde, “Experimental demonstration of nonbilocal quantum correlations,” *Sci. Adv.*, vol. 3, no. 4, p. e1602743, 2017.
- [6] BIG Bell Test Collaboration, “Challenging local realism with human choices.,” *Nature*, vol. 557, no. 7704, p. 212, 2018.

Reviewers' comments:

Reviewer #1 (Remarks to the Author):

I read through the replies to my previous comments and the changes made in the manuscript. I would like to first thank the authors for the detailed replies to my comments. I am satisfied with most of them. However, I have a different opinion on the potential problems caused by post-selecting the events where the external input for measuring a photon in a station coincides with the outcome of measuring another photon at the same station. The authors think that this post-selection is the same as the post-selection due to the low detection efficiency in a Bell test. On the other hand, I think that the uses of external inputs and the corresponding post-selection introduce more fundamental problems than the detection loophole. First, the triangle causal structure studied in this work does not require any external input, while in the experiment reported in this work there are external inputs (although the authors didn't call them external inputs explicitly). How can one justify the experiment as a faithful realization of the triangle causal structure studied? Here I apologize for the unclear statement "this is not a faithful realization of the Fritz distribution" in my previous report. I meant "this is not a faithful realization of the triangle causal structure". Second, the post-selection due to the low detection efficiency can be justified by the fair sampling condition (that is, the sub-ensemble of detected particles is a fair representation of the whole ensemble). Can one justify the post-selection based on the coincidence between the external input and measurement outcome in the same way? It seems to me that the post-selection based on coincidence introduces correlation or artificial causal connection between two events, while the fair sampling conveys the independence of particle behavior from detection efficiency.

To sum up, I don't think that the experiment reported in this work is a faithful realization or demonstration of the triangle casual structure the authors want to study. Therefore I cannot recommend the publication of this work in Nature Communications.

Reviewer #2 (Remarks to the Author):

I have read the revised manuscript, and I happy with all but one of the author's responses to my original comments. I still have some misgivings about the discussion of the source independence.

I appreciate the author's response and essentially agree with what they wrote in the letter. However, I find that their modification of the manuscript does not entirely reflect this. In particular, the following sentence in the paper stands out: "To justify the assumption of source independence, it is essential to use non-synchronized lasers to pump the generation crystals that act as sources."

Of course, using different lasers makes this easier to justify on an intuitive level, but they have not convinced me that it is "essential". Are the lasers powered by the same circuit? Could correlations arise from electrical power supplies? Of course, given our current understanding of the experiment, this is most likely not the case. But it is also most likely not the case, given our current understanding of such experiments, that the same laser pumping three different non-linear crystals will generate correlations between the sources. Perhaps an argument could be made based on the spatial separation of the lasers, but that is not provided.

To be honest, I do not know exactly what they should say about this point. I understand their motivation to use three separate lasers, but it seems to me that this is simply a based on a feeling and not on the underlying physics. I worry that as it is written (worded very strongly, and without appropriate justification) the reader may take the authors' approach as a solution to this problem, when it may not be.

Reviewer #3 (Remarks to the Author):

Regarding my main comment on replacing two of the singlets with classical sources, the authors mention that:

1- This is “a feat rather than a failure of our implementation” and “the fact we have noisy data implies we cannot rely on a standard Bell test”

2- This setup “can in principle be useful also for realizing other quantum distributions” and “in this way, we demonstrate general analysis tools for a real and versatile quantum network, where each involved source is able to generate and distribute quantum states”

In my opinion none of these points are convincing in making the experimental setup complicated. I agree that with this setup one needs new tools, beyond the CHSH inequality, to rule out classicality. Yet as far as I understand the main point of the paper is an experimental demonstration of nonlocality via Fritz’s distribution. If it’s about the the tools to prove nonlocality, as already raised in the reviews, they are more or less known and the argument that no one has applied them on real experimental data is not convincing to me. The authors also argue that this setup (with three shared entangled states) is useful to demonstrate nonlocality of other distributions in the triangle network. This is actually a good point, but why the authors didn’t use this setup for other such distributions in the literature that do require three entangled states as shared sources.

Overall although I believe this paper has several interesting points and has developed techniques that would be useful in the study and experimental realization of network nonlocality, I cannot spot a strong selling point that makes it appropriate for Nature Communications.

Reply to Reviewer 1

I read through the replies to my previous comments and the changes made in the manuscript. I would like to first thank the authors for the detailed replies to my comments. I am satisfied with most of them.

We thank the Reviewer for their careful reading of our work and for acknowledging the appropriateness of our replies to most of the points raised in the previous report. We now resubmit the work after having performed a new experiment that makes use of fast feed-forward of measurements and so without relying on data post selection. In this way, as detailed in what follows below, we fully fix the major issue raised by Reviewer 1.

However, I have a different opinion on the potential problems caused by post-selecting the events where the external input for measuring a photon in a station coincides with the outcome of measuring another photon at the same station. The authors think that this post-selection is the same as the post-selection due to the low detection efficiency in a Bell test. On the other hand, I think that the uses of external inputs and the corresponding post-selection introduce more fundamental problems than the detection loophole. First, the triangle causal structure studied in this work does not require any external input, while in the experiment reported in this work there are external inputs (although the authors didn't call them external inputs explicitly). How can one justify the experiment as a faithful realization of the triangle causal structure studied? Here I apologize for the unclear statement "this is not a faithful realization of the Fritz distribution" in my previous report. I meant "this is not a faithful realization of the triangle causal structure". Second, the post-selection due to the low detection efficiency can be justified by the fair sampling condition (that is, the sub-ensemble of detected particles is a fair representation of the whole ensemble). Can one justify the post-selection based on the coincidence between the external input and measurement outcome in the same way? It seems to me that the post-selection based on coincidence introduces correlation or artificial causal connection between two events, while the fair sampling conveys the independence of particle behavior from detection efficiency.

We thank the Reviewer for stressing this point. We agree that this was a serious drawback from our previous experimental implementation, a post-selected implementation of the Fritz distribution that, as correctly pointed out by the Reviewer, could not be claimed to be a faithful implementation of the triangle causal structure.

Given the relevance of implementing non-classical correlations in the triangle scenario for the very first time, we have decided to adapt and change our experimental setup. In our new experiment, we do not perform this post-selection and indeed implement a faithful realization of the Fritz distribution and the triangle causal structure.

Remember that in order to achieve that, the measurements performed by Alice and Bob correspond to separable measurement on a pair of qubits coming from independent sources, in such a way that the measurement basis of the second photon depends on the measurement outcome of the first. Different from the previous experiment where one could argue that we indeed have two inputs, in this case, the measurements performed by Alice and Bob can be seen as a single POVM measurement (the one described by Fritz in his original paper).

In our new experiment this is achieved by appropriately driving optical switches through an electronic driver which receives signals coming from classical sources Λ_2 and Λ_3 (those connecting Alice and Bob to Charlie) and drives the output port of the optical switch based on the results a_0 and b_0 . The second measurement outcomes, a_1 and b_1 are obtained by performing a polarization measurement on the entangled photons (the state shared by Alice and Bob) produced by the ppKTP source through a half-waveplate (HWP) and a polarizing beam splitter (PBS), implemented in fiber. In turn, the measurement outcomes c_0 and c_1 of Charlie are measured independently by directly feeding the electrical signals produced by Λ_2 and Λ_3 into a time-to-digital converter (TDC).

With this new setup we not only faithfully implement the triangle causal structure but also increased the correlations between a_0 and c_0 as well as between b_0 and c_1 , that is, reduced the potential effect of measurement dependence. In practice, this means that now we are also able to violate a novel class of entropic inequalities that we have recently derived in Ref. [44] of the main text, allowing for an elegant certification of the non-classicality of our triangle network.

To sum up, I don't think that the experiment reported in this work is a faithful realization or demonstration of the triangle casual structure the authors want to study. Therefore I cannot recommend the publication of this work in Nature Communications.

As explained above and detailed in the new version of the manuscript, we have taken the Reviewer's criticism very

seriously and designed a new experimental setup that faithfully implements, for the first time, the Fritz distribution and thus witnesses the non-classicality of correlations arising in a triangle network.

We once more thank the Reviewer for the comment that, we believe, led to substantial improvements in our work and its potential impact. With that, we hope the reviewer can now recommend publication in Nature Communications.

Reply to Reviewer 2

I have read the revised manuscript, and I am happy with all but one of the author's responses to my original comments. I still have some misgivings about the discussion of the source independence.

We thank the Reviewer for the positive assessment of our work and their suggestion. Regarding the point raised by the reviewer, in what follows we provide further clarification.

I appreciate the author's response and essentially agree with what they wrote in the letter. However, I find that their modification of the manuscript does not entirely reflect this. In particular, the following sentence in the paper stands out: "To justify the assumption of source independence, it is essential to use non-synchronized lasers to pump the generation crystals that act as sources." Of course, using different lasers makes this easier to justify on an intuitive level, but they have not convinced me that it is "essential". Are the lasers powered by the same circuit? Could correlations arise from electrical power supplies? Of course, given our current understanding of the experiment, this is most likely not the case. But it is also most likely not the case, given our current understanding of such experiments, that the same laser pumping three different non-linear crystals will generate correlations between the sources. Perhaps an argument could be made based on the spatial separation of the lasers, but that is not provided. To be honest, I do not know exactly what they should say about this point. I understand their motivation to use three separate lasers, but it seems to me that this is simply based on a feeling and not on the underlying physics. I worry that as it is written (worded very strongly, and without appropriate justification) the reader may take the authors' approach as a solution to this problem, when it may not be.

We thank the referee for the comment, which led us to better clarify this point. We first note that, in order to address the post-selection loophole of the previous implementation, we performed a new experiment. Two of the sources have changed, which are now quantum random number generators sharing classically correlated bits among the parties, following the topology of the triangle network. Also, in this case, the independence is enforced by the fact that we use non-synchronized sources whose devices are powered by different electrical outlets.

Regarding the point on the independence of the sources. We notice that the assumption of source independence is mathematically equivalent and physically very similar to the measurement independence assumption (see for instance Ref.[5] of the main text). As such, it is impossible to fully justify it. The best one can do is to make any correlations between the sources as unlikely as possible.

As the Reviewer points out, given our current understanding of such experiments, we would expect that no correlations would be generated even if we use the same laser sources. But that argument would require a precise understanding and description of the physical apparatus, that is, is device dependent. Quite the opposite, our aim here is to work on a device-independent framework and as such only assume constraints arising from the causal structure imposed on the experiment. Think, for instance, of a cryptographic application where the security of the protocol relies on the independence of the sources. In such black box scenario (where we don't know the underlying physics and as such cannot employ the argument of the reviewer) we can be playing against an eavesdropper. Thus, it would make no sense to assume that the correlations coming from a single source (like using the same laser) are in fact independent (that would indeed violate the basis of causal inference, the principle of faithfulness).

In summary, the independence of sources is a loophole in any causal network experiment. The best one can do is to make unfaithful correlations as unlikely as possible and we consider that to use non-synchronized sources is a natural way to achieve that.

In any case, we agree with the Reviewer, and in order to avoid confusions we removed the wording "essential" and added the following sentence in the Experimental Setup section: "Using spatially separated non-synchronized sources, of different natures, enforces the independence of the sources, also having direct applications in quantum communication protocols. Note, however, that the independence of the sources still remains an assumption, considering that this assumption can always be violated by superdeterministic models [73]".

Reply to Reviewer 3

Regarding my main comment on replacing two of the singlets with classical sources, the authors mention that: 1- This is “a feat rather than a failure of our implementation” and “the fact we have noisy data implies we cannot rely on a standard Bell test” 2- This setup “can in principle be useful also for realizing other quantum distributions” and “in this way, we demonstrate general analysis tools for a real and versatile quantum network, where each involved source is able to generate and distribute quantum states” In my opinion, none of these points are convincing in making the experimental setup complicated. I agree that with this setup one needs new tools, beyond the CHSH inequality, to rule out classicality. Yet as far as I understand the main point of the paper is an experimental demonstration of nonlocality via Fritz’s distribution. If it’s about the tools to prove nonlocality, as already raised in the reviews, they are more or less known and the argument that no one has applied them on real experimental data is not convincing to me. The authors also argue that this setup (with three shared entangled states) is useful to demonstrate nonlocality of other distributions in the triangle network. This is actually a good point, but why the authors didn’t use this setup for other such distributions in the literature that do require three entangled states as shared sources.

We thank the reviewer for the comments and questions that certainly have motivated us to perform a new and better experiment implementing, for the first time, the Fritz distribution in a triangle causal structure. In this new version, we perform measurements just as prescribed by Fritz’s original paper (with the feedforward of classical information and real-time basis adaptation, see our Reply to Reviewer 1 for details) and following the Reviewer’s suggestion we have now replaced two of the entangled sources by classical sources of correlation. This has allowed us to significantly improve the correlations between Alice/Bob and Charlie measurement outcomes and add a new and elegant way of testing the non-classicality of the correlations in the triangle network: an entropic inequality that we have recently introduced in [44].

Before entering the details of the new experiment, let us take the chance to answer why we have not used the previous networks with three sources of entangled states to test distribution that indeed needs entanglement in all sources. The answer is simply the unfeasibility of such examples to any experimental test in the near future (at least in photonic platforms). For instance, the distribution found in Ref.[19] of the main text do require entanglement in the three sources, however, it also requires measurements in complicated and unusual entangled basis (that to our knowledge are not known how to be implemented with photons) and furthermore, there is no robustness analysis or any experimentally feasible inequality derived to test it until this day. We certainly agree that entangled measurements are an essential ingredient to unlocking the full power of quantum networks. But one should also notice that we are still at the previous stage: to our knowledge, we are the first to prove experimentally non-classical correlations in a triangle network, without the use of external inputs.

That said, we would like to reinforce that our faithful implementation of the triangle causal structure and Fritz distribution is not a simple old Bell test. There are two complementary ways of seeing it.

First, from a black-box perspective, the question we are addressing in this paper can be cast as: can the correlations observed in our experiment be explained by a classical triangle causal model? Of course, the correlations we generate experimentally are motivated by Bell’s theorem, but our proof of non-classicality is completely independent of it. For instance, anyone accepting that the inflation inequality bounds the set of classically allowed correlations in triangle would agree that a violation of it is a proof of the incompatibility of such correlation with the classical triangle model, even if the person had never heard of Bell’s theorem before (as would be the case for researchers in the field of causal inference).

A second perspective is allowed by the new entropic inequality we now violate experimentally. This inequality relates the level of the CHSH inequality violation with the amount of measurement dependence in a standard Bell test. The point is that while in a standard Bell test one cannot directly access the level of measurement dependence present in the experiment, immersing the Bell causal structure into the triangle that becomes possible. On one side, the correlations between Alice, Bob and Charlie permit to lower bound the amount of dependence the measurement choices of Alice and Bob have with the source shared between them. On the other, a violation of the associated entropic inequality allows us to conclude non-classicality, even if some level of measurement dependence is present in the experiment, something that cannot be done in a standard Bell test. That is, even if one insists on seeing the experimental implementation of the Fritz distribution as a more complicated way of doing a Bell test, that is a novel Bell test, one allowing to quantitatively address the assumption of measurement dependence. Seeing from this perspective, our experiment can thus be seen as complementary to those such as the standard Bell tests with cosmic photons [1] or that employing human randomness (Ref.[41] of the main text). The violation of this entropic inequality is detailed in the section IV.C - “Bounding measurement dependence and violating an entropic inequality for the triangle network” - which we added to the new version of the manuscript.

1. In the Introduction the authors write “... only two examples of photonic networks involving independent sources have been implemented in a device-independent manner so far, ...” It’s unclear why the authors here are limiting their discussion to device-independent networks. The experiment reported here isn’t device independent (in the sense that the usual locality, freedom-of-choice, and detector efficiency loopholes are left open).

The referee is right in saying that “the usual locality, freedom-of-choice, and detector efficiency loopholes are left open”. We indeed stress this points in the main text, briefly in the introduction and more in depth in section III - Experimental Setup. About the locality loophole we state: “In this demonstration, we do not attempt to achieve space-like separation between the registration of the outcomes a, b and c. [...] It is important to note, however, that it would still not justify the lack of a 3-way common cause.” In the same section, we also clarify that: “[...] due to the low efficiencies of the single photon detectors ($\eta \sim 0.5$) we rely on the fair-sampling assumption. [...] closing the detector loophole required decades of effort.” Regarding the freedom-of-choice loophole, we stress in the introduction that in quantum networks and even more so with respect to the triangle network, one can replace the freedom of choice with the source independence assumption, notably “allow the demonstration of nonclassicality without the need of external freely chosen inputs”, as we experimentally show throughout in this work. To justify such assumption in our experimental setup we employ “spatially separated non-synchronized sources, of different natures”, nonetheless noting that “[...]the independence of the sources still remains an assumption since such a loophole cannot be closed in any experiment, considering that this assumption can always be violated by superdeterministic models [73].”. In order to avoid any confusion, in the resubmitted main text, we make all those points clearer and avoided any device independence claim.

Overall although I believe this paper has several interesting points and has developed techniques that would be useful in the study and experimental realization of network nonlocality, I cannot spot a strong selling point that makes it appropriate for Nature Communications.

We thank the Referee for agreeing that our paper does develop relevant techniques. Following the Reviewer’s suggestion, we have improved our experimental results using classical sources and no post-selection. However, we strongly disagree that it lacks the degree of novelty required by a paper in Nature Communications. The triangle network is a paradigmatic scenario that has attracted growing theoretical attention (with several papers in high-impact journals) but lacks any experimental study. How cannot the first experimental test of such a scenario be considered a strong selling point? As a by-product we had to adapt theoretical analysis techniques (data-driven inflation and machine learning), and developed a completely new tool (the entropic inequality recently published in Ref.[44] of the main text) that furthermore allows us to understand our experiment as a Bell test where measurement dependence can be empirically and quantitatively assessed.

The Reviewer’s criticism has undoubtedly improved our experiment, presentation and analysis of the results. We hope to have addressed the main points raised and that now the Reviewer might agree that our work is suitable for publication in Nature Communications.

[1] J. Handsteiner, A. S. Friedman, D. Rauch, J. Gallicchio, B. Liu, H. Hosp, J. Kofler, D. Bricher, M. Fink, C. Leung, *et al.*, “Cosmic Bell test measurement settings from Milky Way stars,” *Phys. Rev. Lett.* **118**, 060401 (2017).

REVIEWER COMMENTS

Reviewer #1 (Remarks to the Author):

Thank the authors for considering my previous comment seriously. I read through the revised manuscript. As far as I can see, the authors have made efforts in order to address my concern; however, one point in the revised experiment is still unclear to me. In the experimental setup depicted in Fig. 2, the classical correlations between Alice and Charlie or between Bob and Charlie are simulated by distributing two random numbers generated by the two QRNGs in the figure. Specifically, the generation of random numbers is entirely completed in the source station labeled Λ_{AC} or Λ_{BC} in the figure. After the random number generation, the same classical signal, i.e., the random number generated, is distributed to Alice and Charlie or Bob and Charlie. I don't think this is a faithful realization of the classically correlated state described in Eq. (6) of the main text. To my understanding, the use of QRNGs here is essentially the same as that in a standard Bell test—Alice and Bob each receive a free input choice. Hence, the revised experiment does not entirely address my concern. The analysis of the experiment reported here can be performed by showing a violation of the standard Bell inequality, in contrast to the authors' claim. For this work to be considered for publication in Nature Communications, the authors must implement the following in their experiment: the classically correlated states described in Eq. (6) should be truly realized in the source stations Λ_{AC} or Λ_{BC} , and these classically correlated states should be distributed and detected at the measurement stations (Alice, Bob, and Charlie) instead of the source stations.

Besides the above, I have a couple of comments on the data analysis techniques used in this work. First, for the standard Bell test, there are techniques for deriving Bell inequalities tailored to the specific data observed in an experiment, see arXiv:0905.2950 and Phys. Rev. A 84, 062118 (2011), for example. Several statements in the manuscript, for instance, the third paragraph in Discussion implies the absence of such techniques. Second, the newly added data analysis based on the violation of entropic inequalities needs more details—the current description of the method in Sect. IV C is too brief. For example, it is hard to understand the last sentence in the caption of Fig. 7. Also, there is no explanation of the meaning of the entropic inequality in Eq. (8). In addition, I would like to point out that Bell violation is robust against measurement dependence, see Phys. Rev. Lett. 113, 190402 (2014) for an illustration. This fact contradicts the last sentence in the first paragraph of Sect. IV C.

Reviewer #2 (Remarks to the Author):

I find the new version of the manuscript, including the new experimental results, quite clear and pedagogical. The author's have completely addressed all of my previous comments.

In the new manuscript, I also find the narrative of the removal of the freedom of choice assumptions from a standard Bell test quite compelling. Hence, I recommend publication.

I have one related question, in this experiment the authors have not closed the locality and detection efficiency loopholes, and they have moved the freedom of choice loophole to a source independence loophole. They discuss the source independence loophole already, and the locality loophole is obvious, but for the detection efficiency how does the threshold in this setup compare to standard Bell tests?

Reviewer #3 (Remarks to the Author):

The authors have properly addressed all my comments and critics. I'm now happy to see this manuscript being published in Nature Communications.

Reply to Reviewer 1

Thank the authors for considering my previous comment seriously. I read through the revised manuscript. As far as I can see, the authors have made efforts in order to address my concern; however, one point in the revised experiment is still unclear to me. In the experimental setup depicted in Fig. 2, the classical correlations between Alice and Charlie or between Bob and Charlie are simulated by distributing two random numbers generated by the two QRNGs in the figure. Specifically, the generation of random numbers is entirely completed in the source station labeled Λ_{AC} or Λ_{BC} in the figure. After the random number generation, the same classical signal, i.e., the random number generated, is distributed to Alice and Charlie or Bob and Charlie. I don't think this is a faithful realization of the classically correlated state described in Eq.(6) of the main text.

We thank the Reviewer for their careful reading of our work and for acknowledging the effort made to address their previous concern.

Regarding the point raised by the Reviewer, in our opinion, it is beyond any doubt the fact that the generated and distributed electrical signals, whose value is ignored until their arrival to Alice and Bob stations, are a perfectly faithful implementation of Eq.(6). More specifically, it is conventional to note that classical shared randomness, which is represented as a nonfactorizing joint probability distribution over a pair of classical variables, if represented in the quantum formalism, is described by a separable state where the local states are mutually orthogonal (i.e., a state of the form of Eq.(6)). Furthermore, a source that distributes classical shared randomness can be used to realize any separable shared state via local operations (classically-controlled local state preparation), so that shared separable quantum states are *causally equivalent* to a source of classical shared randomness. It is in this sense that sharing randomness between Alice and Charlie is, in fact, a faithful realization of the state described in Eq.(6).

In order to make this point clearer, we added in the new manuscript the following clarification (in Experimental Setup section):

To implement the classically correlated sources Λ_{AC} and Λ_{BC} , electrical pulses randomly generated by the shot-noise of distant pairs of single-photon detectors are locally split (boxes labelled Λ_{AC} and Λ_{BC} in Fig. 2); then they are sent to the stations A, C and B, C , respectively, by means of 20m-long electrical cables. Detection of such signals gives values for the bits a_0, b_0, c_0, c_1 . Note that this electrical signal sets up classical correlations (i.e., shared randomness) between Charlie and Alice (Bob), and this is a faithful implementation of the state in Eq.(6).

As stressed in the two points below, the crucial aspect here is the implementation of the triangle structure and the use of the assumption of *this* causal structure in the data analysis. More specifically, realizing the causal arrows, departing from λ_{AC} and λ_{BC} , by means of an electrical signal instead of classical states of photons (generating the same statistics), does not change in any way the conclusions about the nonclassicality of the correlations generated in the experiment. We have no doubt that this is a faithful implementation (the first of its kind) of the triangle causal structure.

To my understanding, the use of QRNGs here is essentially the same as that in a standard Bell test—Alice and Bob each receive a free input choice. Hence, the revised experiment does not entirely address my concern. The analysis of the experiment reported here can be performed by showing a violation of the standard Bell inequality, in contrast to the authors' claim.

The referee claims that the analysis of the experiment reported here can be performed by showing a violation of the standard Bell inequality. As argued below, this is incorrect.

Although we agree that the fact that Alice and Bob each receive a randomly sampled classical variable is common to a Bell experiment and the experiment described in our paper, the key point is that in our data analysis, this randomly sampled variable is treated as an *output* of Alice's lab rather than an input to it, and the parties contend with this by appealing to features of the correlations seen with Charlie and the assumption of the three parties being connected by a triangle causal structure.

To see why the difference between a sampled variable being an input or an output is significant, consider a modification of the Bell experiment wherein the distribution $p(a_1, b_1|a_0, b_0)$ (the one for which Bell inequalities are tested) is such that the variable a_0 , which usually describes an *input* on Alice's side, now describes an *output* on Alice's side, and the variable b_0 , which usually describes an input on Bob's side, now describes an output on Bob's side. For ease of reference, we will call this the "settings-as-outputs" version of a Bell experiment. It is widely acknowledged that in the settings-as-outputs Bell experiment, it is possible to generate distributions $p(a_1, b_1|a_0, b_0)$ that violate Bell inequalities

using local hidden variables. The reason is that if a_0 and b_0 are outputs, then in principle their values could be determined by the hidden variable λ_{AB} and using such a dependence, one can generate *any* distribution $p(a_1, b_1 | a_0, b_0)$ including distributions that violate Bell inequalities (and even distributions that violate the no-signalling condition). In the language of causal modelling, the DAG is simply a fork with Λ_{AB} as the common cause, the composite variable $a = (a_0, a_1)$ as one effect, and the composite variable $b = (b_0, b_1)$ as the other effect, and such a fork is compatible with any joint distribution over a_0, a_1 and b_0, b_1 .

For someone who sought to defend the claim that the “settings-as-outputs” Bell experiment witnesses nonclassicality, it would do no good to insist that although a_0 and b_0 are outputs, one can nonetheless simply *assume* that they are generated by RNGs in Alice and Bob’s labs. In the data analysis, one is not allowed to make such assumptions. The game is to witness nonclassicality assuming only the causal structure of the experiment. In the settings-as-outputs version of the Bell experiment, the causal structure is that of a fork. And there are no Bell-like inequalities for the fork.

If one is to infer that a_0 and b_0 are not determined wholly or in part by Λ_{AB} , then this inference must be based on the observed correlations and the assumed causal structure. In the standard version of the Bell experiment, the assumed causal structure is one wherein a_0 and b_0 are inputs, in which case they do not have Λ_{AB} as a causal parent. As pointed out above, the same inference cannot be made, however, if they are outputs.

Our experimental set-up deviates from the “settings-as-outputs” modification of the Bell experiment because (in the idealized version of our experiment) the variable a_0 that is output at Alice’s wing is *also output* at Charlie’s wing. Call Charlie’s copy a'_0 . The fact that Charlie has a variable a'_0 that is perfectly correlated with a_0 is what proves that a_0 cannot depend in any way on Λ_{AB} . This is the argument that allowed Fritz to prove that there were quantumly realizable distributions in the triangle scenario that could not be realized classically.

The point that we believe the referee is missing is that one needs to distinguish the particular procedures we target in our experiment from *what we assume about those procedures* in the data analysis.

For instance, in the ideal version of the experiment we aim to implement, Charlie does a separable measurement on his pair of systems. But we do not assume such separability in the data analysis, as it would be assuming something about the causal structure internal to Charlie’s lab, and our objective is to witness nonclassicality based only on the assumption of a triangle causal structure *among* the labs.

This point is emphasized in the section entitled “Beyond Bell’s theorem” in our manuscript:

Note that for the triangle scenario, our goal is to witness nonclassicality of the experimentally realized distribution assuming *only* that the causal relations among the three measurement nodes and sources are those described by the triangle scenario. If one were to avail oneself of additional assumptions, in particular, assumptions regarding the causal relations among variables within a given laboratory, then one could witness nonclassicality of our experimental data using standard Bell inequalities. Since such additional assumptions do not hold for all setups that can realize a distribution exhibiting a quantum-classical gap, an analysis which leveraged these additional causal assumptions would not achieve the goal of being applicable to arbitrary data.

Just as we do not assume separability of Charlie’s measurement in the data analysis, neither do we assume that the variable a_0 Alice outputs is a copy of something that was generated at the source station Λ_{AC} . It is in this sense that we could not conclude nonclassicality if the causal structure were that of Bell.

To stress this point even more, we added in the new version of the manuscript, the following paragraph at the end of Sec. II C:

It is worth reiterating here a point made in the beginning of Sec. II, that our goal is to witness nonclassicality using a data analysis technique that assumes *only* the causal structure of the triangle scenario. If we associate a laboratory with each of the nodes in the causal structure, then even though our particular experiment involves specific causal relations between systems *within* the laboratories, the data analysis cannot make use of this extra structure. In other words, we seek a data analysis technique that can witness nonclassicality without assuming any such extra structure. This is the sort of assumption that is appropriate for the device-independent paradigm, wherein the experimental devices are presumed to be supplied by an adversary. All that is presumed to be guaranteed is that the causal relations among the laboratories is the one specified by the triangle scenario. If one *could* avail oneself of the extra structure that is present in the experiment but not part of the description of the triangle scenario, then standard Bell inequalities would be sufficient to witness nonclassicality.

In order to be able to witness the nonclassicality of our data assuming only the triangle causal structure, therefore, we cannot rely on standard Bell inequalities.

And the following footnote within the above text:

For instance, if one could assume that Alice’s output a_0 was a faithful copy of the classical randomness she shares with Charlie and that Bob’s output b_0 was a faithful copy of the classical randomness he shares with Charlie, then one could infer that neither a_0 nor b_0 could depend on Λ_{AB} and consequently having $p(a_1, b_1|a_0, b_0)$ violate a Bell inequality would be sufficient to witness nonclassicality. As a second example, if one could assume that the pair of variables c_0 and c_1 that are outputs of Charlie’s laboratory are such that c_0 depends only on the source shared with Alice and c_1 depends only on the source shared with Bob, then the causal structure being assumed is equivalent to a 4-party line-like structure rather than a triangle scenario. In this case, the full set of Bell inequalities for the conditional distribution $p(a, b|c_0, c_1)$ (where $a = (a_0, a_1)$ and $b = (b_0, b_1)$) are the necessary and sufficient conditions for classicality [1].

For this work to be considered for publication in Nature Communications, the authors must implement the following in their experiment: the classically correlated states described in Eq.(6) should be truly realized in the source stations Λ_{AC} or Λ_{BC} , and these classically correlated states should be distributed and detected at the measurement stations (Alice, Bob, and Charlie) instead of the source stations.

The referee’s claim here seems to be that the variables a_0 and b_0 can only be claimed to be outputs of Alice and Bob’s labs and not inputs (as they are in a standard Bell experiment) if, when we are generating them, we make sure to blind ourselves to their values until they have arrived in Alice’s and Bob’s labs. But in fact, because we make no assumptions in the data analysis about how Alice’s and Bob’s outputs are generated, besides those that are warranted by the causal structure, it is irrelevant whether or not copies of those variables exist at the source station.

The following hopefully clarifies the point. We could imagine that all of the devices used in our experiment were prepared by an adversary. In this version, we cannot trust that there are in fact any RNGs that prepare the state of Eq. 6, nor can we trust that Alice’s output a_0 depends only on the output of some RNG. Maybe the adversary has supplied a device to Alice such that her output a_0 actually depends on Λ_{AB} . Not having any assurances about the devices means that, in the data analysis, one cannot presume anything about how the output a_0 was generated. Our data analysis scheme is of this sort.

Thus, it is irrelevant whether the random numbers are “detected” at the source stations rather than the measurement stations. What is relevant is that *details about how Alice’s output a_0 was generated is no part of the input of the data analysis*. This is what ensures that the data analysis could be done even if the generation scheme was left to an adversary. What we show in the article is that using a data analysis scheme that assumes only that the causal structure is that of the triangle scenario, one can still witness the nonclassicality of the data. The triangle causal structure is distinct from the causal structure of a standard Bell experiment, and so the sort of analysis that is adequate for witnessing nonclassicality under the assumption of a Bell causal structure is not adequate for witnessing nonclassicality under the assumption of a triangle causal structure.

Besides the above, I have a couple of comments on the data analysis techniques used in this work. First, for the standard Bell test, there are techniques for deriving Bell inequalities tailored to the specific data observed in an experiment, see arXiv:0905.2950 and Phys. Rev. A 84, 062118 (2011), for example. Several statements in the manuscript, for instance, the third paragraph in Discussion implies the absence of such techniques.

We certainly agree with the Referee that several data analysis techniques have been developed for the standard Bell test. For instance, the linear programming framework and the statistical data analysis mentioned by the Reviewer.

Through the text, our aim was simply to highlight that this is not the case for networks beyond Bell. In fact, to our knowledge, this is the first time that data-seeded techniques are used by experiments on complex networks where the robust inequalities witnessing non-classicality are extracted from the data.

To avoid any misunderstanding, we now state this explicitly in the Discussion section, where now we also acknowledge the existence of the techniques cited by the Reviewer:

The data analysis techniques we have presented here are also distinguished insofar as they have the capacity to witness nonclassicality for *any* distribution that might arise in an experiment, whereas previous experiments witnessing nonclassicality in novel causal structures beyond Bell have used tools that can only witness the nonclassicality of limited classes of target distributions. This approach thus extends data-seeded techniques previously limited to the standard bipartite Bell scenario [2–5] to the realm of more complex causal networks.

Second, the newly added data analysis based on the violation of entropic inequalities needs more

details—the current description of the method in Sect. IV C is too brief. For example, it is hard to understand the last sentence in the caption of Fig. 7. Also, there is no explanation of the meaning of the entropic inequality in Eq. (8).

Following the Reviewer’s recommendation we have given further details about the entropic inequality violated in the experiment (we refer to the work [44] of the new manuscript for details). In particular, we added the following paragraph in Sec. IV C:

In this modified Bell scenario, shown in Fig.7(b), one can lower bound the measurement dependence, quantified via the mutual information $I(\lambda_{AB} : a_0, b_0)$ between the source λ_{AB} and the measurement settings a_0 and b_0 , relating it with the violation of the CHSH inequality [6, 7]. Further, employing the entropic approach [8–11], this mutual information can also be upper bounded by an entropic function that involves only observable variables and so can be extracted directly from the experimental data. Combining both the upper and lower bounds on $I(\lambda_{AB} : a_0, b_0)$, one arrives at a Bell inequality blending probabilities and entropies, the violation of which witnesses the nonclassicality of the data, irrespectively of any potential measurement dependence $I(\lambda_{AB} : a_0, b_0)$ present in the experiment. This inequality is given by (see Ref.[12] for the further details)

$$\mathcal{E} \equiv 2 - S^{CHSH} + \sqrt{\frac{16 \Theta(a_0, b_0, C)}{\log_2 e}} \geq 0, \quad (1)$$

where S^{CHSH} is the standard CHSH quantity evaluated on $P(a_1 b_1 | a_0 b_0)$ [13], and

$$\Theta(a_0, b_0, C) := \min \begin{cases} H(a_0, b_0 | C), \\ H(a_0, b_0) - I(a_0 : b_0 : C) - I(a_0 : C) - I(b_0 : C), \\ H(a_0, b_0) + H(C) - 2I(a_0 : b_0 : C) - 2I(a_0 : C) - 2I(b_0 : C), \end{cases} \quad (2)$$

with $I(a_0 : b_0 : C) := H(a_0, b_0, C) - H(a_0, b_0) - H(a_0, C) - H(b_0, C) + H(a_0) + H(b_0) + H(C)$ the tripartite mutual information and $H(X) = -\sum_x p(x) \log p(x)$ the Shannon entropy relative to the variable X .

We also modify the Caption of Fig.7 as follows:

Triangle scenario from extended Bell scenario. (a) Extended Bell scenario with measurement dependence. Relative to the standard Bell scenario, the source λ_{AB} is presumed to influence not only the outcomes a_1 and b_1 , but the setting variables a_0 and b_0 as well. This allows for measurement dependence and can describe superdeterministic models [14]. (b) Extended Bell causal structure with measurement dependence mapped into the triangle scenario. Relative to the standard Bell scenario, one posits an additional laboratory, associated to Charlie, an additional source λ_{AC} between Alice and Charlie and an additional source λ_{BC} between Bob and Charlie. Correlations between a_0 and c_0, c_1 imply an upper bound on the potential dependence of a_0 on λ_{AB} , described by the entropic inequality in Eq. (8). Similarly, correlations between b_0 and c_0, c_1 imply an upper bound on the potential dependence of b_0 on λ_{AB} .

In addition, I would like to point out that Bell violation is robust against measurement dependence, see Phys. Rev. Lett. 113, 190402 (2014) for an illustration. This fact contradicts the last sentence in the first paragraph of Sect. IV C.

The Reviewer is right when stating that, in principle, a Bell inequality violation can be made robust against some degree of measurement dependence. However, crucially, in a Bell scenario where the causal structure is relaxed, allowing for causal influences between the source and the measurement setting, some amount of measurement independence has to be *assumed* in order to witness nonclassicality from data. Indeed, if one looks only at the relaxed causal structure and no prior restriction is imposed on the measurement dependence, then any violation of a Bell inequality can be explained by classical local models (see Brans, International Journal of Theoretical Physics (1988)).

It is also worth remarking that the robustness results presented in Phys. Rev. Lett. 113, 190402 (2014) relies on a very specific quantifier of measurement dependence. This is in contrast, for instance, with the results in Phys. Rev. Lett. 105, 250404 (2010), showing that a very small amount of measurement dependence (quantified in a similar manner to what we do in our paper, via the mutual information between the hidden variable and the inputs) is already enough to simulate all quantum violations of the CHSH inequality.

In any case, as demonstrated in [44], in a triangle scenario we can bound solely from the data and in a device-independent way, the degree of measurement dependence present in the experiment. That is, contrary to the Bell case, this amount of measurement dependence does not need to be assumed. The causal structure itself allows witnessing the nonclassicality of the statistics, a task that is impossible in a standard Bell scenario.

To make this point clearer we added in the first paragraph of Sec. IV the following footnote:

In this modification, any amount of measurement dependence is in principle allowed between the hidden variable and the measurement settings. Consequently, even though the scenario is related to Bell's, the nonclassicality exhibited necessarily goes beyond that which one finds in Bell's scenario because in the latter measurement dependence allows for a classical account of *any* correlations ⁴.

⋮

⁴ In a Bell scenario where causal influences between the source and the measurement settings are allowed, some amount of measurement independence has to be *assumed* in order to witness nonclassicality from the data [60,78,79], otherwise, any violation of a Bell inequality can be explained by classical local models [80].

Reply to Reviewer 2

I find the new version of the manuscript, including the new experimental results, quite clear and pedagogical. The author's have completely addressed all of my previous comments. In the new manuscript, I also find the narrative of the removal of the freedom of choice assumptions from a standard Bell test quite compelling. Hence, I recomemnd publication.

We thank the Reviewer for the positive assessment of our work with their recommendation for publication in Nature Communication and for all the relevant suggestions that certainly have helped us to improve our results and their presentation.

I have one related question, in this experiment the authors have not closed the locality and detection efficiency loopholes, and they have moved the freedom of choice loophole to a source independence loophole. They discuss the source independence loophole already, and the locality loophole is obvious, but for the detection efficiency how does the threshold in this setup compare to standard Bell tests?

The question on the detection efficiency threshold in the triangle scenario is certainly very timely and interesting. However, to our knowledge, contrary to the standard Bell scenario, the threshold detection efficiencies for causal networks (in particular the triangle) have not been studied in the literature. Due to the non-convexity of the set of correlations, this is a highly non-trivial question that deserves detailed future investigation. We now stress this point in Sec. III of the manuscript, where we say:

Furthermore, due to the low efficiencies of the single photon detectors ($\eta \sim 0.5$) and the fact that the threshold values required for closing the detector loophole in the triangle scenario are not yet known, we rely on the fair-sampling assumption.

Reply to Reviewer 3

The authors have properly addressed all my comments and critics. I'm now happy to see this manuscript being published in Nature Communications.

We thank the reviewer for the positive assessment of the new manuscript and experiment and for recommending the publication in Nature Communications. We thank the Reviewer once more for the suggestions and remarks that have substantially improved our paper.

-
- [1] R. J. Evans, "Graphs for margins of Bayesian networks," *Scandinavian J. Stat.* **43**, 625 (2016).
 - [2] N. Brunner, D. Cavalcanti, S. Pironio, V. Scarani, and S. Wehner, "Bell nonlocality," *Rev. Mod. Phys.* **86**, 419 (2014).
 - [3] V. Scarani, *Bell nonlocality* (Oxford University Press, 2019).
 - [4] M. B. Elliott, "A linear program for testing local realism," arXiv preprint arXiv:0905.2950 (2009).
 - [5] Y. Zhang, S. Glancy, and E. Knill, "Asymptotically optimal data analysis for rejecting local realism," *Physical Review A* **84**, 062118 (2011).
 - [6] R. Chaves, R. Kueng, J. B. Brask, and D. Gross, "Unifying framework for relaxations of the causal assumptions in Bell's theorem," *Phys. Rev. Lett.* **114**, 140403 (2015).
 - [7] M. J. Hall and C. Branciard, "Measurement-dependence cost for Bell nonlocality Causal versus retrocausal models," *Phys. Rev. A* **102**, 052228 (2020).
 - [8] R. Chaves, L. Luft, T. O. Maciel, D. Gross, D. Janzing, and B. Schölkopf, "Inferring latent structures via information inequalities," arXiv:1407.2256 (2014).
 - [9] T. Fritz and R. Chaves, "Entropic inequalities and marginal problems," *IEEE Trans. Info. Theo.* **59**, 803 (2012).
 - [10] R. Chaves, L. Luft, and D. Gross, "Causal structures from entropic information geometry and novel scenarios," *New J. Phys.* **16**, 043001 (2014).
 - [11] R. Chaves, C. Majenz, and D. Gross, "Information-theoretic implications of quantum causal structures," *Nature comm.* **6**, 1 (2015).
 - [12] R. Chaves, G. Moreno, E. Polino, D. Poderini, I. Agresti, A. Suprano, M. R. Barros, G. Carvacho, E. Wolfe, A. Canabarro, *et al.*, "Causal networks and freedom of choice in bell's theorem," *PRX Quantum* **2**, 040323 (2021).
 - [13] J. F. Clauser, M. A. Horne, A. Shimony, and R. A. Holt, "Proposed experiment to test local hidden-variable theories," *Phys. Rev. Lett.* **23**, 880 (1969).
 - [14] S. Hossenfelder and T. Palmer, "Rethinking superdeterminism," *Frontiers in Physics* **8**, 139 (2020).

REVIEWERS' COMMENTS

Reviewer #1 (Remarks to the Author):

I read through the replies to my previous comments and the changes made in the manuscript. I am satisfied with and convinced by most of them. However, I still have a different opinion on whether the triangle causal structure studied in this work automatically guarantees that a_0 (b_0) is an output of Alice (Bob). In the worst scenario, the current experimental implementation might introduce a loophole, but the main conclusion of this work does not change. Hence, I would like to recommend the publication of the current manuscript in Nature Communications.

Reply to Reviewer 1

I read through the replies to my previous comments and the changes made in the manuscript. I am satisfied with and convinced by most of them. However, I still have a different opinion on whether the triangle causal structure studied in this work automatically guarantees that a_0 (b_0) is an output of Alice (Bob). In the worst scenario, the current experimental implementation might introduce a loophole, but the main conclusion of this work does not change. Hence, I would like to recommend the publication of the current manuscript in Nature Communications.

We thank the Reviewer for their careful reading of our work and for recommending its publication in Nature Communications. We agree with the Reviewer that the different opinion on the experimental implementation does not change the main conclusion of this work.